# THE DEEP BOOTSTRAP FRAMEWORK: GOOD ONLINE LEARNERS ARE GOOD OFFLINE GENERALIZERS

**Preetum Nakkiran**
Harvard University*
preetum@cs.harvard.edu

**Behnam Neyshabur**
Blueshift, Alphabet
neyshabur@google.com

**Hanie Sedghi**
Google Research, Brain team
hsedghi@google.com

## ABSTRACT

We propose a new framework for reasoning about generalization in deep learning. The core idea is to couple the Real World, where optimizers take stochastic gradient steps on the empirical loss, to an Ideal World, where optimizers take steps on the population loss. This leads to an alternate decomposition of test error into: (1) the Ideal World test error plus (2) the gap between the two worlds. If the gap (2) is universally small, this reduces the problem of generalization in offline learning to the problem of optimization in online learning. We then give empirical evidence that this gap between worlds can be small in realistic deep learning settings, in particular supervised image classification. For example, CNNs generalize better than MLPs on image distributions in the Real World, but this is "because" they optimize faster on the population loss in the Ideal World. This suggests our framework is a useful tool for understanding generalization in deep learning, and lays a foundation for future research in the area.

## 1 INTRODUCTION

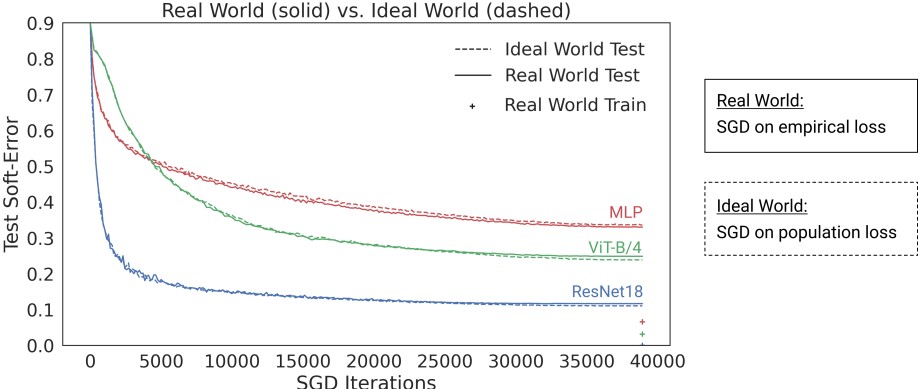

Figure 1: Three architectures trained from scratch on CIFAR-5m, a CIFAR-10-like task. The Real World is trained on 50K samples for 100 epochs, while the Ideal World is trained on 5M samples in 1 pass. The Real World Test remains close to Ideal World Test, despite a large generalization gap.

The goal of a generalization theory in supervised learning is to understand when and why trained models have small test error. The classical framework of generalization decomposes the test error of a model $f_t$ as:

$$\text{TestError}(f_t) = \text{TrainError}(f_t) + \underbrace{\left[\text{TestError}(f_t) - \text{TrainError}(f_t)\right]}_{\text{Generalization gap}} \tag{1}$$

and studies each part separately (e.g. Vapnik and Chervonenkis (1971); Blumer et al. (1989); Shalev-Shwartz and Ben-David (2014)). Many works have applied this framework to study generalization of deep networks (e.g. Bartlett (1997); Bartlett et al. (1999); Bartlett and Mendelson (2002); Anthony and Bartlett (2009); Neyshabur et al. (2015b); Dziugaite and Roy (2017); Bartlett et al. (2017); Neyshabur et al. (2017); Harvey et al. (2017); Golowich et al. (2018); Arora et al. (2018; 2019);

Allen-Zhu et al. (2019); Long and Sedghi (2019); Wei and Ma (2019)). However, there are at least two obstacles to understanding generalization of modern neural networks via the classical approach.

1. Modern methods can *interpolate*, reaching TrainError $\approx 0$, while still performing well. In these settings, the decomposition of Equation (1) does not actually reduce test error into two different subproblems: it amounts to writing TestError = 0 + TestError. That is, understanding the generalization gap here is exactly equivalent to understanding the test error itself.
2. Most if not all techniques for understanding the generalization gap (e.g. uniform convergence, VC-dimension, regularization, stability, margins) remain vacuous (Zhang et al., 2017; Belkin et al., 2018a;b; Nagarajan and Kolter, 2019) and not predictive (Nagarajan and Kolter, 2019; Jiang et al., 2019; Dziugaite et al., 2020) for modern networks.

In this work, we propose an alternate approach to understanding generalization to help overcome these obstacles. The key idea is to consider an alternate decomposition:

$$\text{TestError}(f_t) = \underbrace{\text{TestError}(f_t^{\text{iid}})}_{\text{A: Online Learning}} + \underbrace{[\text{TestError}(f_t) - \text{TestError}(f_t^{\text{iid}})]}_{\text{B: Bootstrap error}} \quad (2)$$

where $f_t$ is the neural-network after $t$ optimization steps (the "Real World"), and $f_t^{\text{iid}}$ is a network trained identically to $f_t$, but using fresh samples from the distribution in each mini-batch step (the "Ideal World"). That is, $f_t^{\text{iid}}$ is the result of optimizing on the *population loss* for $t$ steps, while $f_t$ is the result of optimizing on the *empirical loss* as usual (we define this more formally later).

This leads to a different decoupling of concerns, and proposes an alternate research agenda to understand generalization. To understand generalization in the bootstrap framework, it is sufficient to understand:

(A) **Online Learning:** How quickly models optimize on the population loss, in the infinite-data regime (the Ideal World).
(B) **Finite-Sample Deviations:** How closely models behave in the finite-data vs. infinite-data regime (the bootstrap error).

Although neither of these points are theoretically understood for deep networks, they are closely related to rich areas in optimization and statistics, whose tools have not been brought fully to bear on the problem of generalization. The first part (A) is purely a question in online stochastic optimization: We have access to a stochastic gradient oracle for a population loss function, and we are interested in how quickly an online optimization algorithm (e.g. SGD, Adam) reaches small population loss. This problem is well-studied in the online learning literature for convex functions (Bubeck, 2011; Hazan, 2019; Shalev-Shwartz et al., 2011), and is an active area of research in non-convex settings (Jin et al., 2017; Lee et al., 2016; Jain and Kar, 2017; Gao et al., 2018; Yang et al., 2018; Maillard and Munos, 2010). In the context of neural networks, optimization is usually studied on the empirical loss landscape (Arora et al., 2019; Allen-Zhu et al., 2019), but we propose studying optimization on the population loss landscape directly. This highlights a key difference in our approach: we never compare test and train quantities— we only consider test quantities.

The second part (B) involves approximating fresh samples with "reused" samples, and reasoning about behavior of certain functions under this approximation. This is closely related to the *nonparametric bootstrap* in statistics (Efron, 1979; Efron and Tibshirani, 1986), where sampling from the population distribution is approximated by sampling with replacement from an empirical distribution. Bootstrapped estimators are widely used in applied statistics, and their theoretical properties are known in certain cases (e.g. Hastie et al. (2009); James et al. (2013); Efron and Hastie (2016); Van der Vaart (2000)). Although current bootstrap theory does not apply to neural networks, it is conceivable that these tools could eventually be extended to our setting.

**Experimental Validation.** Beyond the theoretical motivation, our main experimental claim is that the bootstrap decomposition is actually useful: in realistic settings, the bootstrap error is often small, and the performance of real classifiers is largely captured by their performance in the Ideal World. Figure 1 shows one example of this, as a preview of our more extensive experiments in Section 4. We plot the test error of a ResNet (He et al., 2016a), an MLP, and a Vision Transformer (Dosovitskiy et al., 2020) on a CIFAR-10-like task, over increasing minibatch SGD iterations. The Real World is trained on 50K samples for 100 epochs. The Ideal World is trained on 5 million samples with a single pass. Notice that the bootstrap error is small for all architectures, although the generalization

gap can be large. In particular, the convnet generalizes better than the MLP on finite data, but this is "because" it optimizes faster on the population loss with infinite data. See Appendix D.1 for details.

**Our Contributions.**

- **Framework:** We propose the Deep Bootstrap framework for understanding generalization in deep learning, which connects offline generalization to online optimization. (Section 2).
- **Validation:** We give evidence that the bootstrap error is small in realistic settings for supervised image classification, by conducting extensive experiments on large-scale tasks (including variants of CIFAR-10 and ImageNet) for many architectures (Section 4). Thus,

    *The generalization of models is largely determined by*
    *their optimization speed in online and offline learning.*

- **Implications:** We highlight how our framework can unify and yield insight into important phenomena in deep learning, including implicit bias, model selection, data-augmentation and pretraining (Section 5). In particular:

    *Good models and training procedures are those which*
    *(1) optimize quickly in the Ideal World and*
    *(2) do not optimize too quickly in the Real World.*

**Additional Related Work.** The bootstrap error is also related to algorithmic stability (e.g. Bousquet and Elisseeff (2001); Hardt et al. (2016)), since both quantities involve replacing samples with fresh samples. However, stability-based generalization bounds cannot tightly bound the bootstrap error, since there are many settings where the generalization gap is high, but bootstrap error is low.

## 2  THE DEEP BOOSTRAP

Here we more formally describe the Deep Bootstrap framework and our main claims. Let $\mathcal{F}$ denote a learning algorithm, including architecture and optimizer. We consider optimizers which can be used in online learning, such as stochastic gradient descent and variants. Let $\mathrm{Train}_{\mathcal{F}}(\mathcal{D}, n, t)$ denote training in the "Real World": using the architecture and optimizer specified by $\mathcal{F}$, on a train set of $n$ samples from distribution $\mathcal{D}$, for $t$ optimizer steps. Let $\mathrm{Train}_{\mathcal{F}}(\mathcal{D}, \infty, t)$ denote this same optimizer operating on the population loss (the "Ideal World"). Note that these two procedures use identical architectures, learning-rate schedules, mini-batch size, etc – the only difference is, the Ideal World optimizer sees a fresh minibatch of samples in each optimization step, while the Real World reuses samples in minibatches. Let the Real and Ideal World trained models be:

$$\text{Real World:} \quad f_t \leftarrow \mathrm{Train}_{\mathcal{F}}(\mathcal{D}, n, t)$$

$$\text{Ideal World:} \quad f_t^{\mathrm{iid}} \leftarrow \mathrm{Train}_{\mathcal{F}}(\mathcal{D}, \infty, t)$$

We now claim that for all $t$ until the Real World converges, these two models $f_t, f_t^{\mathrm{iid}}$ have similar test performance. In our main claims, we differ slightly from the presentation in the Introduction in that we consider the "soft-error" of classifiers instead of their hard-errors. The soft-accuracy of classifiers is defined as the softmax probability on the correct label, and (soft-error) := 1 − (soft-accuracy). Equivalently, this is the expected error of temperature-1 samples from the softmax distribution. Formally, define $\varepsilon$ as the bootstrap error – the gap in soft-error between Real and Ideal worlds at time $t$:

$$\mathrm{TestSoftError}_{\mathcal{D}}(f_t) = \mathrm{TestSoftError}_{\mathcal{D}}(f_t^{\mathrm{iid}}) + \varepsilon(n, \mathcal{D}, \mathcal{F}, t) \tag{3}$$

Our main experimental claim is that the bootstrap error $\varepsilon$ is uniformly small in realistic settings.

**Claim 1 (Bootstrap Error Bound, informal)** *For choices of $(n, \mathcal{D}, \mathcal{F})$ corresponding to realistic settings in deep learning for supervised image classification, the bootstrap error $\varepsilon(n, \mathcal{D}, \mathcal{F}, t)$ is small for all $t \leq T_0$. The "stopping time" $T_0$ is defined as the time when the Real World reaches small training error (we use 1%) – that is, when Real World training has essentially converged.*

The restriction on $t \leq T_0$ is necessary, since as $t \to \infty$ the Ideal World will continue to improve, but the Real World will at some point essentially stop changing (when train error $\approx 0$). However, we claim that these worlds are close for "as long as we can hope"— as long as the Real World optimizer is still moving significantly.

**Error vs. Soft-Error.** We chose to measure soft-error instead of hard-error in our framework for both empirical and theoretically-motivated reasons. Empirically, we found that the bootstrap gap is often smaller with respect to soft-errors. Theoretically, we want to define the bootstrap gap such that it converges to 0 as data and model size are scaled to infinity. Specifically, if we consider an *overparameterized* scaling limit where the Real World models always interpolate the train data, then Distributional Generalization (Nakkiran and Bansal, 2020) implies that the bootstrap gap for test error will *not* converge to 0 on distributions with non-zero Bayes risk. Roughly, this is because the Ideal World classifier will converge to the Bayes optimal one ($\mathrm{argmax}_y\, p(y|x)$), while the Real World interpolating classifier will converge to a *sampler* from $p(y|x)$. Considering soft-errors instead of errors nullifies this issue. We elaborate further on the differences between the worlds in Section 6. See also Appendix C for relations to the nonparametric bootstrap (Efron, 1979).

## 3 EXPERIMENTAL SETUP

Our bootstrap framework could apply to any setting where an iterative optimizer for online learning is applied in an offline setting. In this work we primarily consider stochastic gradient descent (SGD) applied to deep neural networks for image classification. This setting is well-developed both in practice and in theory, and thus serves as an appropriate first step to vet theories of generalization, as done in many recent works (e.g. Jiang et al. (2019); Neyshabur et al. (2018); Zhang et al. (2017); Arora et al. (2019)). Our work does not depend on overparameterization— it holds for both under and over parameterized networks, though it is perhaps most interesting in the overparameterized setting. We now describe our datasets and experimental methodology.

### 3.1 DATASETS

Measuring the bootstrap error in realistic settings presents some challenges, since we do not have enough samples to instantiate the Ideal World. For example, for a Real World CIFAR-10 network trained on 50K samples for 100 epochs, the corresponding Ideal World training would require 5 million samples (fresh samples in each epoch). Since we do not have 5 million samples for CIFAR-10, we use the following datasets as proxies. More details, including sample images, in Appendix E.

**CIFAR-5m.** We construct a dataset of 6 million synthetic CIFAR-10-like images, by sampling from the CIFAR-10 Denoising Diffusion generative model of Ho et al. (2020), and labeling the unconditional samples by a 98.5% accurate Big-Transfer model (Kolesnikov et al., 2019). These are synthetic images, but close to CIFAR-10 for research purposes. For example, a WideResNet28-10 trained on 50K samples from CIFAR-5m reaches 91.2% test accuracy on CIFAR-10 test set. We use 5 million images for training, and reserve the rest for the test set. We plan to release this dataset.

**ImageNet-DogBird.** To test our framework in more complex settings, with real images, we construct a distribution based on ImageNet ILSVRC-2012 (Russakovsky et al., 2015). Recall that we need a setting with a large number of samples relative to the difficulty of the task: if the Real World performs well with few samples and few epochs, then we can simulate it in the Ideal World. Thus, we construct a simpler binary classification task out of ImageNet by collapsing classes into the superclasses "hunting dog" and "bird." This is a roughly balanced task with 155K total images.

### 3.2 METHODOLOGY

For experiments on CIFAR-5m, we exactly simulate the Real and Ideal worlds as described in Section 2. That is, for every Real World architecture and optimizer we consider, we construct the corresponding Ideal World by executing the exact same training code, but using fresh samples in each epoch. The rest of the training procedure remains identical, including data-augmentation and learning-rate schedule. For experiments on ImageNet-DogBird, we do not have enough samples to exactly simulate the Ideal World. Instead, we approximate the Ideal World by using the full training set ($N = 155K$) and data-augmentation. Formally, this corresponds to approximating $\mathrm{Train}_{\mathcal{F}}(\mathcal{D}, \infty, t)$ by $\mathrm{Train}_{\mathcal{F}}(\mathcal{D}, 155K, t)$. In practice, we train the Real World on $n = 10K$ samples for 120 epochs, so we can approximate this with less than 8 epochs on the full $155K$ train set. Since we train with data augmentation (crop+resize+flip), each of the 8 repetitions of each sample will undergo different random augmentations, and thus this plausibly approximates fresh samples.

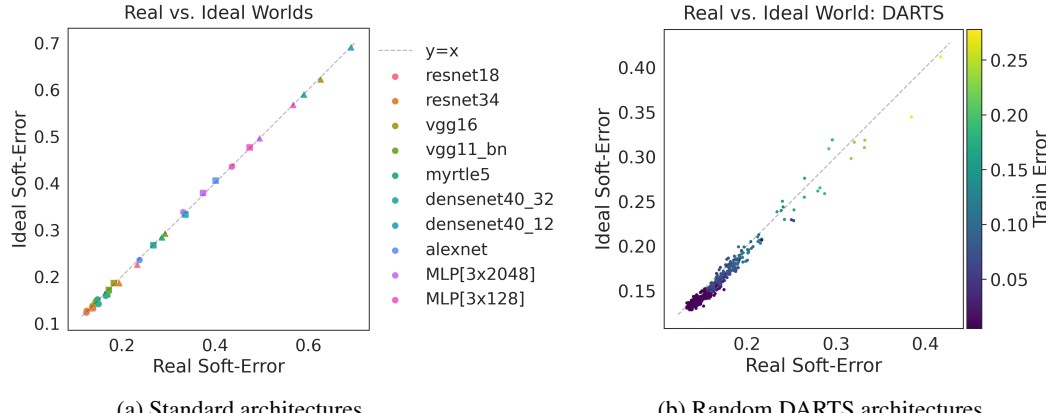

(a) Standard architectures.
(b) Random DARTS architectures.

Figure 2: **Real vs Ideal World: CIFAR-5m.** SGD with 50K samples. (a): Varying learning-rates $0.1 (\bullet), 0.01 (\blacksquare), 0.001 (\blacktriangle)$. (b): Random architectures from DARTS space (Liu et al., 2019).

**Stopping time.** We stop both Real and Ideal World training when the Real World reaches a small value of train error (which we set as $1\%$ in all experiments). This stopping condition is necessary, as described in Section 2. Thus, for experiments which report test performance "at the end of training", this refers to either when the target number of epochs is reached, or when Real World training has converged ($< 1\%$ train error). We always compare Real and Ideal Worlds after the exact same number of train iterations.

## 4 MAIN EXPERIMENTS

We now give evidence to support our main experimental claim, that the bootstrap error $\varepsilon$ is often small for realistic settings in deep learning for image classification. In all experiments in this section, we instantiate the same model and training procedure in the Real and Ideal Worlds, and observe that the test soft-error is close at the end of training. Full experimental details are in Appendix D.2.

**CIFAR-5m.** In Figure 2a we consider a variety of standard architectures on CIFAR-5m, from fully-connected nets to modern convnets. In the Real World, we train these architectures with SGD on $n = 50K$ samples from CIFAR-5m, for 100 total epochs, with varying initial learning rates. We then construct the corresponding Ideal Worlds for each architecture and learning rate, trained in the same way with fresh samples each epoch. Figure 2a shows the test soft-error of the trained classifiers in the Real and Ideal Worlds at the end of training. Observe that test performance is very close in Real and Ideal worlds, although the Ideal World sees $100\times$ unique samples during training.

To test our framework for more diverse architectures, we also sample 500 random architectures from the DARTS search space (Liu et al., 2019). These are deep convnets of varying width and depth, which range in size from 70k to 5.5 million parameters. Figure 2b shows the Real and Ideal World test performance at the end of training— these are often within $3\%$.

**ImageNet: DogBird.** We now test various ImageNet architectures on ImageNet-DogBird. The Real World models are trained with SGD on $n = 10K$ samples with standard ImageNet data augmentation. We approximate the Ideal World by training on $155K$ samples as described in Section 3.2. Figure 3a plots the Real vs. Ideal World test error at the end of training, for various architectures. Figure 3b shows this for ResNet-18s of varying widths.

## 5 DEEP PHENOMENA THROUGH THE BOOTSTRAP LENS

Here we show that our Deep Bootstrap framework can be insightful to study phenomena and design choices in deep learning. For example, many effects in the Real World can be seen through their corresponding effects in the Ideal World. Full details for experiments provided in Appendix D.

**Model Selection in the Over- and Under-parameterized Regimes.** Much of theoretical work in deep learning focuses on overparameterized networks, which are large enough to fit their train sets.

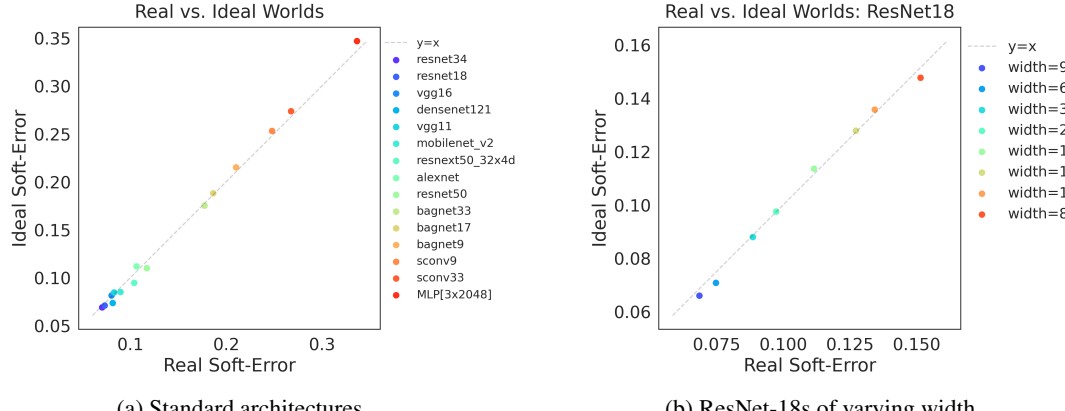

(a) Standard architectures.

(b) ResNet-18s of varying width.

Figure 3: **ImageNet-DogBird.** Real World models trained on 10K samples.

However, in modern practice, state-of-the-art networks can be either over or *under*-parameterized, depending on the scale of data. For example, SOTA models on 300 million JFT images or 1 billion Instagram images are underfitting, due to the massive size of the train set (Sun et al., 2017; Mahajan et al., 2018). In NLP, modern models such as GPT-3 and T5 are trained on massive internet-text datasets, and so are solidly in the underparameterized regime (Kaplan et al., 2020; Brown et al., 2020; Raffel et al., 2019). We highlight one surprising aspect of this situation:

*The same techniques (architectures and training methods)*
*are used in practice in both over- and under-parameterized regimes.*

For example, ResNet-101 is competitive both on 1 billion images of Instagram (when it is underparameterized) and on 50k images of CIFAR-10 (when it is overparameterized). This observation was made recently in Bornschein et al. (2020) for overparameterized architectures, and is also consistent with the conclusions of Rosenfeld et al. (2019). It is apriori surprising that the same architectures do well in both over and underparameterized regimes, since there are very different considerations in each regime. In the overparameterized regime, architecture matters for generalization reasons: there are many ways to fit the train set, and some architectures lead SGD to minima that generalize better. In the underparameterized regime, architecture matters for purely optimization reasons: all models will have small generalization gap with 1 billion+ samples, but we seek models which are capable of reaching low values of test loss, and which do so quickly (with few optimization steps). Thus, it should be surprising that in practice, we use similar architectures in both regimes.

Our work suggests that these phenomena are closely related: If the boostrap error is small, then we should expect that architectures which optimize well in the infinite-data (underparameterized) regime also generalize well in the finite-data (overparameterized) regime. This unifies the two apriori different principles guiding model-selection in over and under-parameterized regimes, and helps understand why the same architectures are used in both regimes.

**Implicit Bias via Explicit Optimization.** Much recent theoretical work has focused on the *implicit bias* of gradient descent (e.g. Neyshabur et al. (2015a); Soudry et al. (2018); Gunasekar et al. (2018b;a); Ji and Telgarsky (2019); Chizat and Bach (2020)). For overparameterized networks, there are many minima of the empirical loss, some which have low test error and others which have high test error. Thus studying why interpolating networks generalize amounts to studying why SGD is "biased" towards finding empirical minima with low population loss. Our framework suggests an alternate perspective: instead of directly trying to characterize which empirical minima SGD reaches, it may be sufficient to study why SGD optimizes quickly on the population loss. That is, instead of studying implicit bias of optimization on the empirical loss, we could study explicit properties of optimization on the population loss.

The following experiment highlights this approach. Consider the D-CONV and D-FC architectures introduced recently by Neyshabur (2020). D-CONV is a deep convolutional network and D-FC is its fully-connected counterpart: an MLP which subsumes the convnet in expressive capacity. That is, D-FC is capable of representing all functions that D-CONV can represent, since it replaces all conv layers with fully-connected layers and unties all the weights. Both networks reach close to 0 train

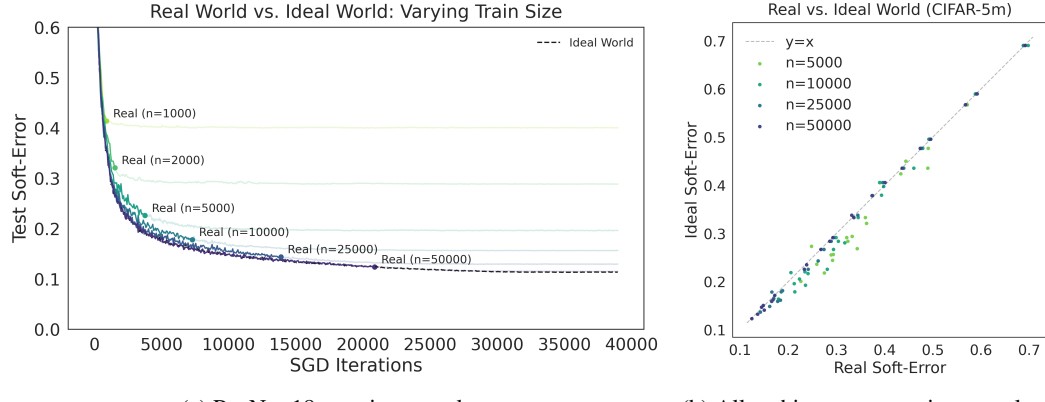

(a) ResNet-18, varying samples.      (b) All architectures, varying samples.

Figure 4: **Effect of Sample Size.**

error on 50K samples from CIFAR-5m, but the convnet generalizes much better. The traditional explanation for this is that the "implicit bias" of SGD biases the convnet to a better-generalizing minima than the MLP. We show that, in fact, this generalization is captured by the fact that D-CONV optimizes much faster on the population loss than D-FC. Figure 5c shows the test and train errors of both networks when trained on 50K samples from CIFAR-5m, in the Real and Ideal Worlds. Observe that the Real and Ideal world test performances are nearly identical.

**Sample Size.** In Figure 4, we consider the effect of varying the train set size in the Real World. Note that in this case, the Ideal World does not change. There are two effects of increasing $n$: (1) The stopping time extends— Real World training continues for longer before converging. And (2) the bootstrap error decreases. Of these, (1) is the dominant effect. Figure 4a illustrates this behavior in detail by considering a single model: ResNet-18 on CIFAR-5m. We plot the Ideal World behavior of ResNet-18, as well as different Real Worlds for varying $n$. All Real Worlds are stopped when they reach $< 1\%$ train error, as we do throughout this work. After this point their test performance is essentially flat (shown as faded lines). However, until this stopping point, all Real Worlds are roughly close to the Ideal World, becoming closer with larger $n$. These learning curves are representative of most architectures in our experiments. Figure 4b shows the same architectures of Figure 2a, trained on various sizes of train sets from CIFAR-5m. The Real and Ideal worlds may deviate from each other at small $n$, but become close for realistically large values of $n$.

**Data Augmentation.** Data augmentation in the Ideal World corresponds to randomly augmenting each fresh sample before training on it (as opposed to re-using the same sample for multiple augmentations). There are 3 potential effects of data augmentation in our framework: (1) it can affect the Ideal World optimization, (2) it can affect the bootstrap gap, and (3) it can affect the Real World stopping time (time until training converges). We find that the dominant factors are (1) and (3), though data augmentation does typically reduce the bootstrap gap as well. Figure 5a shows the effect of data augmentation on ResNet-18 for CIFAR-5m. In this case, data augmentation does not change the Ideal World much, but it extends the time until the Real World training converges. This view suggests that good data augmentations should (1) not hurt optimization in the Ideal World (i.e., not destroy true samples much), and (2) obstruct optimization in the Real World (so the Real World can improve for longer before converging). This is aligned with the "affinity" and "diversity" view of data augmentations in Gontijo-Lopes et al. (2020). See Appendix B.3 for more figures, including examples where data augmentation hurts the Ideal World.

**Pretraining.** Figure 5b shows the effect of pretraining for Image-GPT (Chen et al., 2020), a transformer pretrained for generative modeling on ImageNet. We fine-tune iGPT-S on 2K samples of CIFAR-10 (not CIFAR-5m, since we have enough samples in this case) and compare initializing from an early checkpoint vs. the final pretrained model. The fully-pretrained model generalizes better in the Real World, and also optimizes faster in the Ideal World. Additional experiments including ImageNet-pretrained Vision Transformers (Dosovitskiy et al., 2020) are in Appendix B.5.

**Random Labels.** Our approach of comparing Real and Ideal worlds also captures generalization in the random-label experiment of Zhang et al. (2017). Specifically, if we train on a distribution with purely random labels, both Real and Ideal world models will have trivial test error.

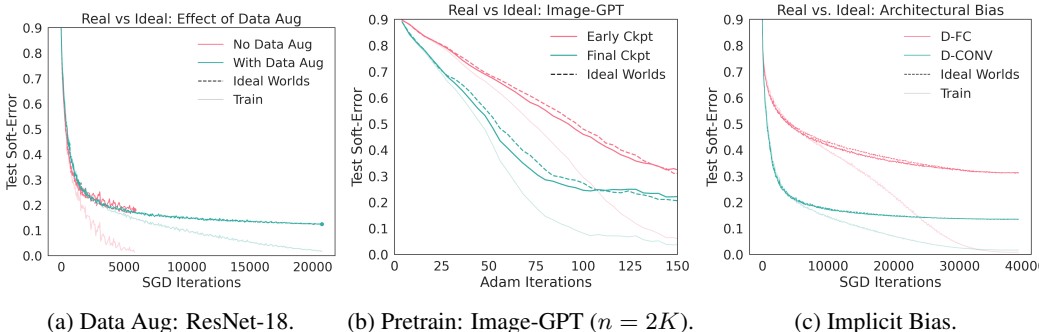

(a) Data Aug: ResNet-18.  (b) Pretrain: Image-GPT ($n = 2K$).  (c) Implicit Bias.

Figure 5: **Deep Phenomena in Real vs. Ideal Worlds.**

## 6 DIFFERENCES BETWEEN THE WORLDS

In our framework, we only compare the test soft-error of models in the Real and Ideal worlds. We do not claim these models are close in all respects— in fact, this is not true. For example, Figure 6 shows the same ResNet-18s trained in the Introduction (Figure 1), measuring three different metrics in both worlds. Notably, the test *loss* diverges drastically between the Real and Ideal worlds, although the test soft-error (and to a lesser extent, test error) remains close. This is because training to convergence in the Real World will cause the network weights to grow unboundedly, and the softmax distribution to concentrate (on both train and test). In contrast, training in the Ideal World will generally not cause weights to diverge, and the softmax will remain diffuse. This phenomenon also means that the Error and Soft-Error are close in the Real World, but can be slightly different in the Ideal World, which is consistent with our experiments.

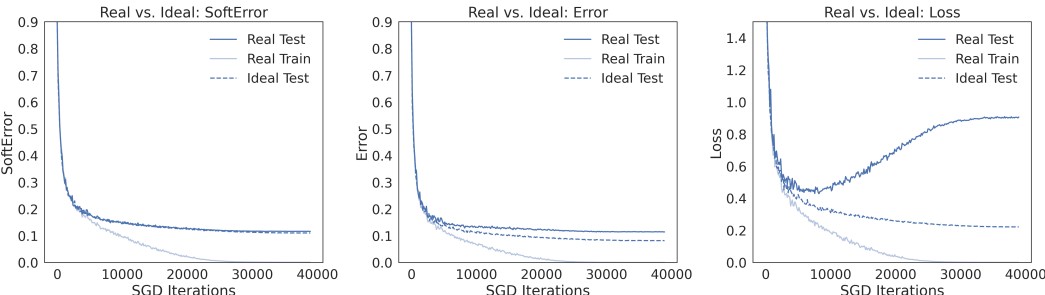

Figure 6: **SoftError vs. Error vs. Loss: ResNet-18.**

## 7 CONCLUSION AND DISCUSSION

We propose the Deep Bootstrap framework for understanding generalization in deep learning. Our approach is to compare the Real World, where optimizers take steps on the empirical loss, to an Ideal World, where optimizers have infinite data and take steps on the population loss. We find that in modern settings, the test performance of models is close between these worlds. This establishes a new connection between the fields of generalization and online learning: models which learn quickly (online) also generalize well (offline). Our framework thus provides a new lens on deep phenomena, and lays a promising route towards theoretically understanding generalization in deep learning.

**Limitations.** Our work takes first steps towards characterizing the bootstrap error $\varepsilon$, but fully understanding this, including its dependence on problem parameters $(n, \mathcal{D}, \mathcal{F}, t)$, is an important area for future study. The bootstrap error is not universally small for all models and learning tasks: for example, we found the gap was larger at limited sample sizes and without data augmentation. Moreover, it can be large in simple settings like linear regression (Appendix A), or settings when the Real World test error is non-monotonic (e.g. due to epoch double-decent (Nakkiran et al., 2020)). Nevertheless, the gap appears to be small in realistic deep learning settings, and we hope that future work can help understand why.

ACKNOWLEDGEMENTS

Work completed in part while PN was interning at Google. PN also supported in part by a Google PhD Fellowship, the Simons Investigator Awards of Boaz Barak and Madhu Sudan, and NSF Awards under grants CCF 1565264, CCF 1715187 and IIS 1409097.

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

## A    TOY EXAMPLE

Here we present a theoretically-inspired toy example, giving a simple setting where the bootstrap gap is small, but the generalization gap is large. We also give an analogous example where the bootstrap error is large. The purpose of these examples is (1) to present a simple setting where the bootstrap framework can be more useful than studying the generalization gap. And (2) to illustrate that the bootstrap gap is not always small, and can be large in certain standard settings.

We consider the following setup. Let us pass to a regression setting, where we have a distribution over $(x, y) \in \mathbb{R}^d \times \mathbb{R}$, and we care about mean-square-error instead of classification error. That is, for a model $f$, we have $\text{TestMSE}(f) := \mathbb{E}_{x,y}[(f(x) - y)^2]$. Both our examples are from the following class of distributions in dimension $d = 1000$.

$$x \sim \mathcal{N}(0, V)$$
$$y := \sigma(\langle \beta^*, x \rangle)$$

where $\beta^* \in \mathbb{R}^d$ is the ground-truth, and $\sigma$ is a pointwise activation function. The model family is linear,

$$f_\beta(x) := \langle \beta, x \rangle$$

We draw $n$ samples from the distribution, and train the model $f_\beta$ using full-batch gradient descent on the empirical loss:

$$\text{TrainMSE}(f_\beta) := \frac{1}{n} \sum_i (f(x_i) - y_i)^2 = \frac{1}{n} ||X\beta - y||^2$$

We chose $\beta^* = e_1$, and covariance $V$ to be diagonal with 10 eigenvalues of 1 and the remaining eigenvalues of 0.1. That is, $x$ is essentially 10-dimensional, with the remaining coordinates "noise."

The two distributions are instances of the above setting for different choices of parameters.

- **Setting A.** Linear activation $\sigma(x) = x$. With $n = 20$ train samples.
- **Setting B.** Sign activation $\sigma(x) = \text{sgn}(x)$. With $n = 100$ train samples.

Setting A is a standard well-specified linear regression setting. Setting B is a misspecified regression setting. Figure 7 shows the Real and Ideal worlds in these settings, for gradient-descent on the empirical loss (with step-size $\eta = 0.1$). Observe that in the well-specified Setting A, the Ideal World performs much better than the Real World, and the bootstrap framework is not as useful. However, in the misspecified Setting B, the bootstrap gap remains small even as the generalization gap grows.

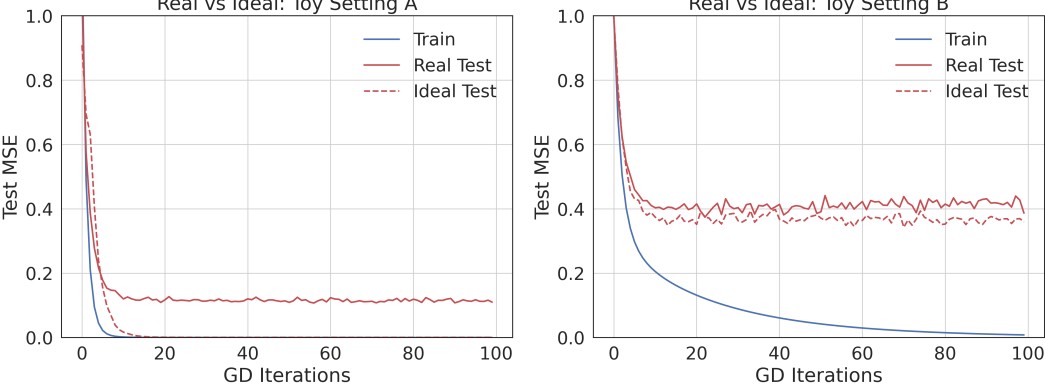

Figure 7: **Toy Example.** Examples of settings with large and small bootstrap error.

This toy example is contrived to help isolate factors important in more realistic settings. We have observed behavior similar to Setting B in other simple settings with real data, such as regression on MNIST/Fashion-MNIST, as well as in the more complex settings in the body of this paper.

# B    ADDITIONAL FIGURES

## B.1    INTRODUCTION EXPERIMENT

Figure 8 shows the same experiment as Figure 1 in the Introduction, including the train error in the Real World. Notice that the bootstrap error remains small, even as the generalization gap (between train and test) grows.

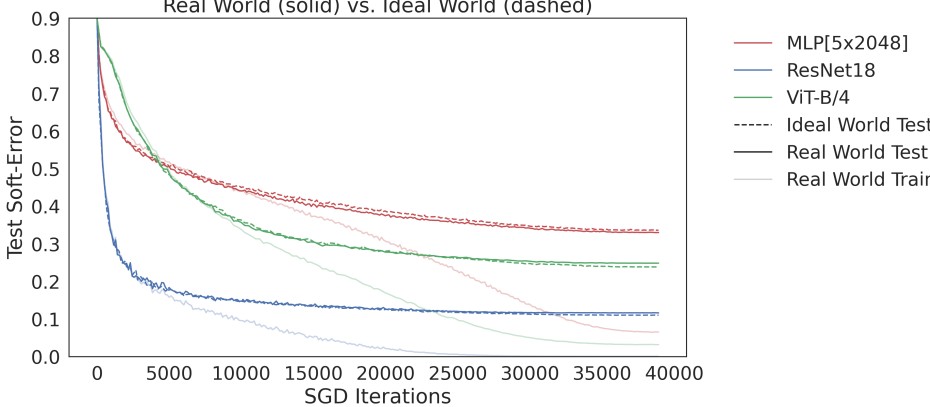

Figure 8: The corresponding train soft-errors for Figure 1.

## B.2    DARTS ARCHITECTURES

Figure 9 shows the Real vs Ideal world for trained random DARTS architectures.

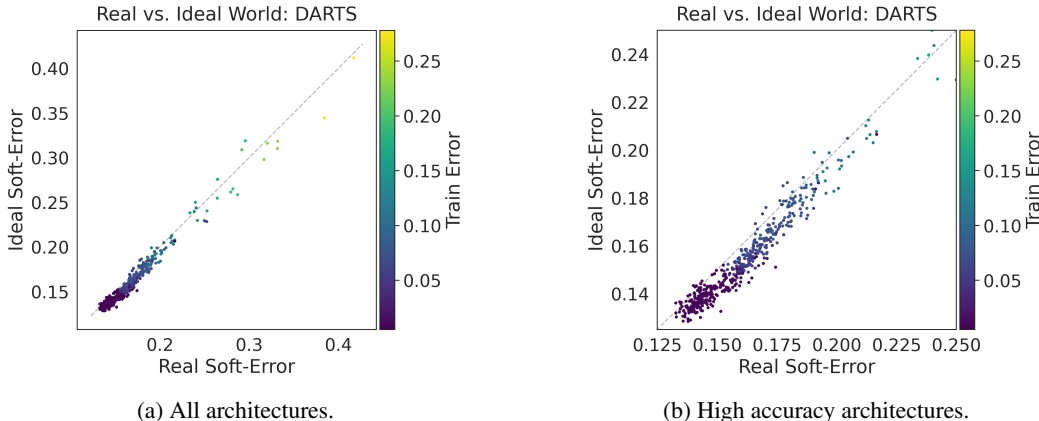

(a) All architectures.                                    (b) High accuracy architectures.

Figure 9: **Random DARTS Architectures.** Panel (b) shows zoomed view of panel (a).

## B.3    EFFECT OF DATA AUGMENTATION

Figure 10 shows the effect of data-augmentation in the Ideal World, for several selected architectures on CIFAR-5m. Recall that data augmentation in the Ideal World corresponds to randomly augmenting each fresh sample once, as opposed to augmenting the same sample multiple times. We train with SGD using the same hyperparameters as the main experiments (described in Appendix D.2). We use standard CIFAR-10 data augmentation: random crop and random horizontal flip.

The test performance without augmentation is shown as solid lines, and with augmentation as dashed lines. Note that VGG and ResNet do not behave differently with augmentation, but augmentation significantly hurts AlexNet and the MLP. This may be because VGG and ResNet have global spatial pooling, which makes them (partially) shift-invariant, and thus more amenable to the random cropping. In contrast, augmentation hurts the architectures without global pooling, perhaps because for these architectures, augmented samples appear more out-of-distribution.

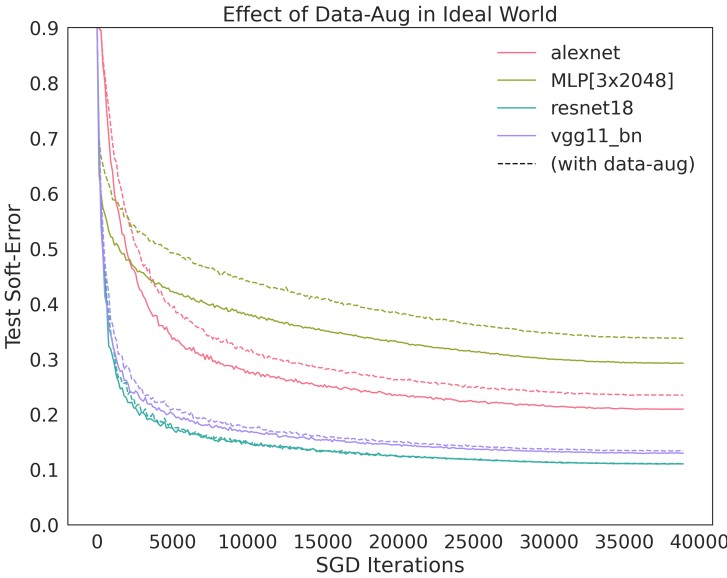

Figure 10: Effect of Data Augmentation in the Ideal World.

Figure 11a shows the same architectures and setting as Figure 2 but trained without data augmentation. That is, we train on 50K samples from CIFAR-5m, using SGD with cosine decay and initial learning rate $\{0.1, 0.01, 0.001\}$.

Figure 11b shows learning curves with and without data augmentation of a ResNet-18 on $n = 10k$ samples. This is the analogous setting of Figure 5a in the body, which is for $n = 50k$ samples.

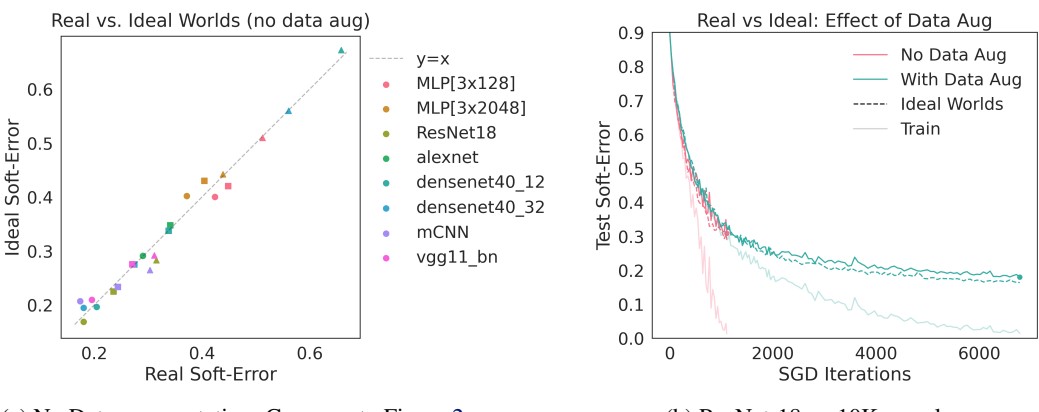

(a) No Data-augmentation. Compare to Figure 2.

(b) ResNet-18 on 10K samples.

Figure 11: **Effect of Data Augmentation.**

## B.4 ADAM

Figure 12 shows several experiments with the Adam optimizer (Kingma and Ba, 2014) in place of SGD. We train all architectures on 50K samples from CIFAR-5m, with data-augmentation, batchsize 128, using Adam with default parameters (lr=0.001, $\beta_1 = 0.9, \beta_2 = 0.999$).

## B.5 PRETRAINING

### B.5.1 PRETRAINED MLP

Figure 14 shows the effect of pretraining for an MLP (3x2048) on CIFAR-5m, by comparing training from scratch (random initialization) to training from an ImageNet-pretrained initialization. The pretrained MLP generalizes better in the Real World, and also optimizes faster in the Ideal World. We fine tune on 50K samples from CIFAR-5m, with no data-augmentation.

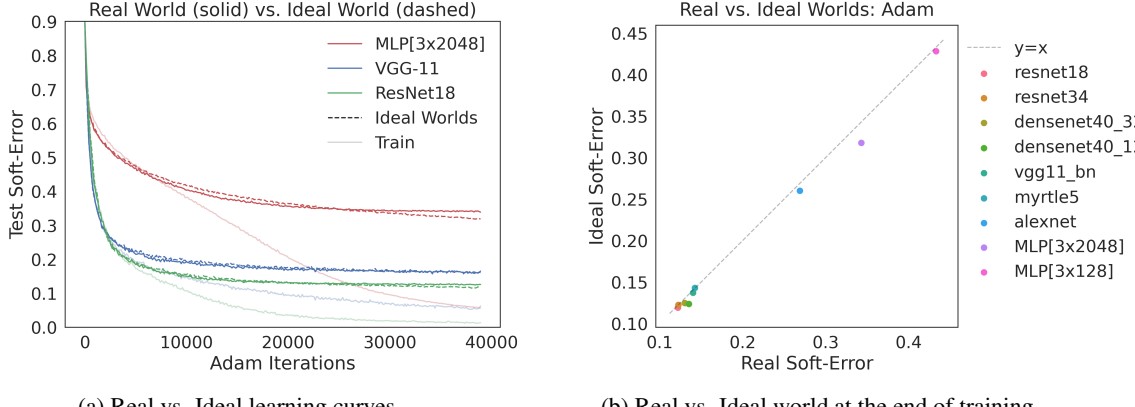

(a) Real vs. Ideal learning curves.

(b) Real vs. Ideal world at the end of training.

Figure 12: **Adam Experiments.** For various architectures on 50K samples from CIFAR-5m.

For ImageNet-pretraining, we train the MLP[3x2048] on full ImageNet (224px, 1000 classes), using Adam with default settings, and batchsize 1024. We use standard ImageNet data augmentation (random resized crop + horizontal flip) and train for 500 epochs. This MLP achieves test accuracy 21% and train accuracy 30% on ImageNet. For fine-tuning, we adapt the network to 32px input size by resizing the first layer filters from 224x224 to 32x32 via bilinear interpolation. We then replace the classification layer, and fine-tune the entire network on CIFAR-5m.

### B.5.2 PRETRAINED VISION TRANSFORMER

Figure 13 shows the effect of pretraining for Vision Transformer (ViT-B/4). We compare ViT-B/4 trained from scratch to training from an ImageNet-pretrained initialization. The color of line in Figure 13 indicates the pretraining strategy, and the weight of the line indicates the measurment (Ideal World test error, Real World test error, or Real World train error). We fine tune on 50K samples from CIFAR-5m, with standard CIFAR-10 data augmentation. Notice that pretrained ViT generalizes better in the Real World, and also optimizes correspondingly faster in the Ideal World.

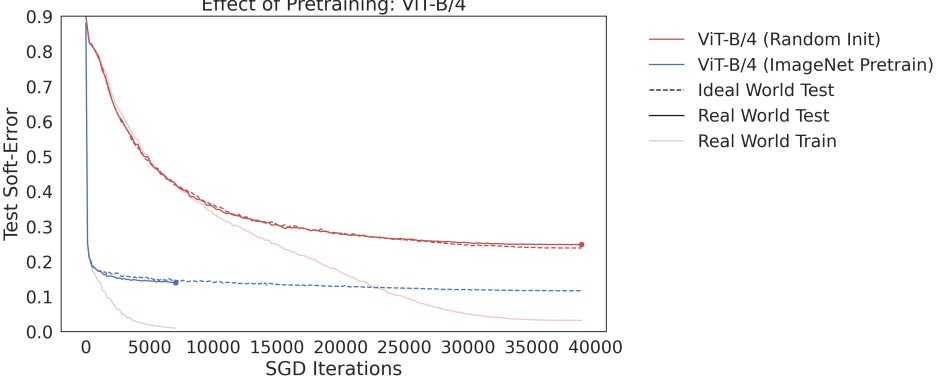

Figure 13: Real vs. Ideal Worlds for Vision Transformer on CIFAR-5m, with and w/o pretraining.

Both ViT models are fine-tuned using SGD identical to Figure 1 in the Introduction, as described in Section D.1. For ImageNet pretraining, we train on ImageNet resized to $32 \times 32$, after standard data augmentation. We pretrain for 30 epochs using Adam with batchsize 2048 and constant learning rate 1e-4. We then replace and zero-initialize the final layer in the MLP head, and fine-tune the full model for classification on CIFAR-5m. This pretraining process it not as extensive as in Dosovitskiy et al. (2020); we use it to demonstrate that our framework captures the effect of pretraining in various settings.

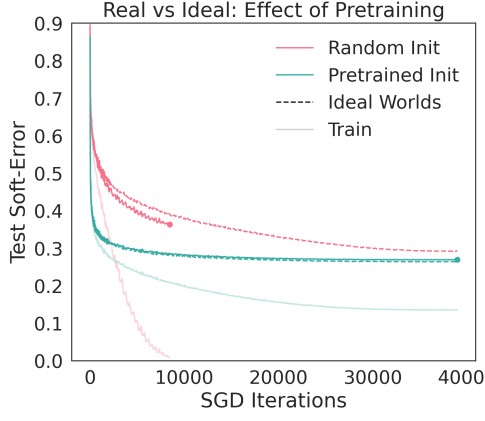 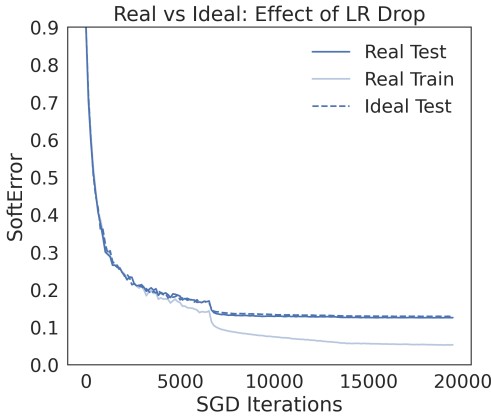

Figure 14: ImageNet-Pretraining: MLP[3x2048].  Figure 15: Effect of Learning Rate Drop.

### B.6 LEARNING RATE

Figure 2a shows that the Real and Ideal world remain close across varying initial learning rates. All of the figures in the body use a cosine decay learning rate schedule, but this is only for simplicity; we observed that the effect of various learning rate schedules are mirrored in Real and Ideal worlds. For example, Figure 15 shows a ResNet18 in the Real World trained with SGD for 50 epochs on CIFAR-5m, with a step-wise decay schedule (initial LR 0.1, dropping by factor 10 at $1/3$ and $2/3$ through training). Notice that the Ideal World error drops correspondingly with the Real World, suggesting that the LR drop has a similar effect on the population optimization as it does on the empirical optimization.

### B.7 DIFFERENCE BETWEEN WORLDS

Figure 16 shows test soft-error, test error, and test loss for the MLP from Figure 1.

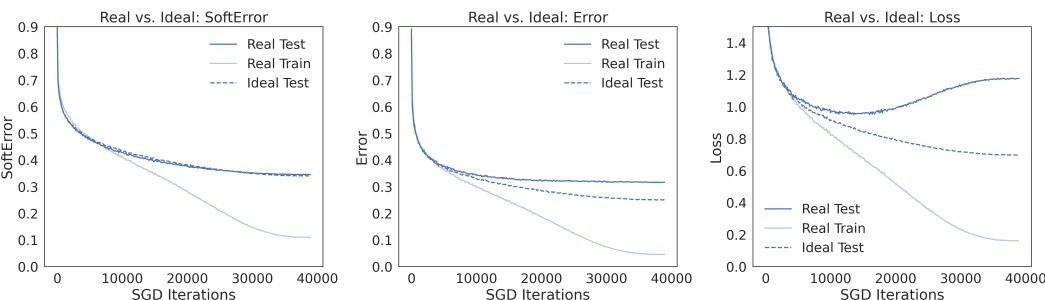

Figure 16: **SoftError vs. Error vs. Loss: MLP[5x2048].**

### B.8 ERROR VS. SOFTERROR

Here we show the results of several of our experiments if we measure the bootstrap gap with respect to Test Error instead of SoftError. The bootstrap gap is often reasonably small even with respect to Error, though it is not as well behaved as SoftError.

Figure 17 shows the same setting as Figure 2a in the body, but measuring Error instaed of SoftError.

#### B.8.1 TRAINING WITH MSE

We can measure Test Error even for networks which do not naturally output a probability distribution. Here, we train various architectures on CIFAR-5m using the squared-loss (MSE) directly on logits, with no softmax layer. This follows the methodology in Hui and Belkin (2020). We train all Real-World models using SGD, batchsize 128, momentum 0.9, initial learning rate 0.002 with cosine decay for 100 epochs.

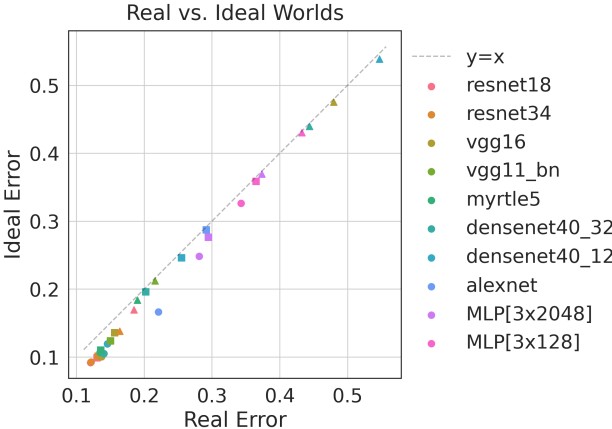

Figure 17: Measuring Test Error instead of SoftError. Compare to Figure 2a

Figure 18 shows the Test Error and Test Loss in the Real and Ideal Worlds. The bootstrap gap, with respect to test error, for MSE-trained networks is reasonably small – though there are deviations in the low error regime. Compare this to Figure 2a, which measures the SoftError for networks trained with cross-entropy.

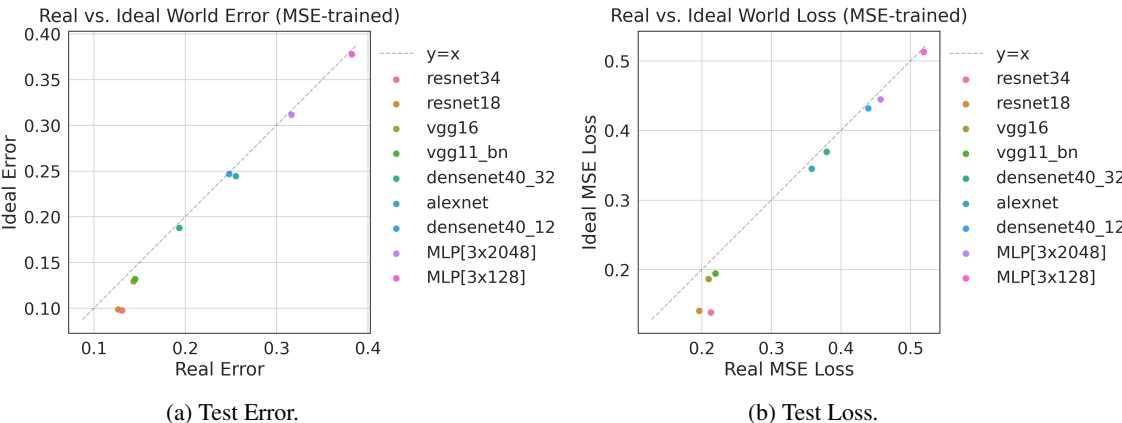

(a) Test Error.  (b) Test Loss.

Figure 18: **Real vs. Ideal: Training with Squared Loss.**

## C  BOOTSTRAP CONNECTION

Here we briefly describe the connection between our Deep Bootstrap framework and the nonparametric bootstrap of Efron (1979).

For an online learning procedure $\mathcal{F}$ and a sequence of labeled samples $\{x_i\}$, let $\mathrm{Train}^{\mathcal{F}}(x_1, x_2, \dots)$ denote the function which optimizes on the samples $x_1, x_2, \dots$ in sequence, and outputs the resulting model. (For example, the function which initializes a network of a certain architecture, and takes successive gradient steps on the sequence of samples, and outputs the resulting model).

For a given $(n, \mathcal{D}, \mathcal{F}, t)$, define the function $G : \mathcal{X}^t \to \mathbb{R}$ as follows. $G$ takes as input $t$ labeled samples $\{x_i\}$, and outputs the Test Soft-Error (w.r.t $\mathcal{D}$) of training on the sequence $\{x_i\}$. That is,

$$G(x_1, x_2, \dots x_t) := \mathrm{TestSoftError}_{\mathcal{D}}(\mathrm{Train}^{\mathcal{F}}(x_1, x_2, \dots, x_t))$$

Now, the Ideal World test error is simply $G$ evaluated on iid samples $x_i \sim \mathcal{D}$:

$$\text{Ideal World:} \quad \mathrm{TestSoftError}_{\mathcal{D}}(f_t^{\mathrm{iid}}) = G(\{x_i\}) \quad \text{where } x_i \sim \mathcal{D}$$

The Real World, using a train set of size $n < t$, is equivalent[1] to evaluating $G$ on $t$ examples sampled with replacement from a train set of size $n$. This corresponds to training on the same sample multiple times, for $t$ total train steps.

$$\text{Real World:} \quad \mathrm{TestSoftError}_{\mathcal{D}}(f_t) = G(\{\widetilde{x}_i\}) \quad \text{where } S \sim \mathcal{D}^n; \widetilde{x}_i \sim S$$

Here, the samples $\widetilde{x}_i$ are drawn with replacement from the train set $S$. Thus, the Deep Bootstrap error $\varepsilon = G(\{\widetilde{x}_i\}) - G(\{x_i\})$ measures the deviation of a certain function when it is evaluated on iid samples v.s. on samples-with-replacement, which is exactly the form of bootstrap error in applications of the nonparametric bootstrap (Efron, 1979; Efron and Tibshirani, 1986; 1994).

---

[1]Technically we do not sample-with-replacement in the experiments, we simply reuse each sample a fixed number of times (once in each epoch). We describe it as sampling-with-replacement here to more clearly relate it to the nonparametric bootstrap.

## D    APPENDIX: EXPERIMENTAL DETAILS

**Technologies.** All experiments run on NVIDIA V100 GPUs. We used PyTorch (Paszke et al., 2019), NumPy (Harris et al., 2020), Hugging Face transformers (Wolf et al., 2019), pandas (McKinney et al., 2010), W&B (Biewald, 2020), Matplotlib (Hunter, 2007), and Plotly (Inc., 2015).

### D.1    INTRODUCTION EXPERIMENT

All architectures in the Real World are trained with $n = 50K$ samples from CIFAR-5m, using SGD on the cross-entropy loss, with cosine learning rate decay, for 100 epochs. We use standard CIFAR-10 data augmentation of random crop+horizontal flip. All models use batch size 128, so they see the same number of samples at each point in training.

The ResNet is a preactivation ResNet18 (He et al., 2016b), the MLP has 5 hidden layers of width 2048, with pre-activation batch norm. The Vision Transformer uses the ViT-Base configuration from Dosovitskiy et al. (2020), with a patch size of $4 \times 4$ (adapted for the smaller CIFAR-10 image size of $32 \times 32$). We use the implementation from `https://github.com/lucidrains/vit-pytorch`. We train all architectures including ViT from scratch, with no pretraining. ResNets and MLP use initial learning rate 0.1 and momentum 0.9. ViT uses initial LR 0.01, momentum 0.9, and weight decay 1e-4. We did not optimize ViT hyperparameters as extensively as in Dosovitskiy et al. (2020); this experiment is only to demonstrate that our framework is meaningful for diverse architectures.

Figure 1 plots the Test Soft-Error over the course of training, and the Train Soft-Error at the end of training. We plot median over 10 trials (with random sampling of the train set, random initialization, and random SGD order in each trial).

### D.2    MAIN EXPERIMENTS

For CIFAR-5m we use the following architectures: AlexNet (Krizhevsky et al., 2012), VGG (Simonyan and Zisserman, 2015), Preactivation ResNets (He et al., 2016b), DenseNet (Huang et al., 2017). The Myrtle5 architecture is a 5-layer CNN introduced by (Page, 2018).

In the Real World, we train these architectures on $n = 50K$ samples from CIFAR-5m using cross-entropy loss. All models are trained with SGD with batchsize 128, initial learning rate $\{0.1, 0.01, 0.001\}$, cosine learning rate decay, for 100 total epochs, with data augmentation: random horizontal flip and `RandomCrop(32, padding=4)`. We plot median over 10 trials (with random sampling of the train set, random initialization, and random SGD order in each trial).

**DARTS Architectures.** We sample architectures from the DARTS search space (Liu et al., 2019), as implemented in the codebase of Dong and Yang (2020). We follow the parameters used for CIFAR-10 in Dong and Yang (2020), while also varying width and depth for added diversity. Specifically, we use 4 nodes, number of cells $\in \{1, 5\}$, and width $\in \{16, 64\}$. We train all DARTS architectures with SGD, batchsize 128, initial learning rate 0.1, cosine learning rate decay, for 100 total epochs, with standard augmentation (random crop+flip).

**ImageNet: DogBird** All architectures for ImageNet-DogBird are trained with SGD, batchsize 128, learning rate 0.01, momentum 0.9, for 120 epochs, with standard ImageNet data augmentation (random resized crop to 224px, horizontal flip). We report medians over 10 trials for each architecture.

We additional include the ImageNet architectures: BagNet (Brendel and Bethge, 2019), MobileNet (Sandler et al., 2018), and ResNeXt (Xie et al., 2017). The architectures SCONV9 and SCONV33 refer to the S-CONV architectures defined by Neyshabur (2020), instantiated for ImageNet with base-width 48, image size 224, and kernel size $\{9, 33\}$ respectively.

### D.3    IMPLICIT BIAS

We use the D-CONV architecture from (Neyshabur, 2020), with base width 32, and the corresponding D-FC architecture. PyTorch specification of these architectures are provided in Appendix F for convenience. We train both architectures with SGD, batchsize 128, initial learning rate 0.1, cosine

learning rate decay, for 100 total epochs, with random crop + horizontal flip data-augmentation. We plot median errors over 10 trials.

## D.4 IMAGE-GPT FINETUNING

We fine-tune iGPT-S, using the publicly available pretrained model checkpoints from Chen et al. (2020). The "Early" checkpoint in Figure 5b refers to checkpoint 131000, and the "Final" checkpoint is 1000000. Following Chen et al. (2020), we use Adam with ($lr = 0.003, \beta_1 = 0.9, \beta_2 = 0.95$), and batchsize 128. We do not use data augmentation. For simplicity, we differ slightly from Chen et al. (2020) in that we simply attach the classification head to the [average-pooled] last transformer layer, and we fine-tune using only classification loss and not the joint generative+classification loss used in Chen et al. (2020). Note that we fine-tune the entire model, not just the classification head.

# E APPENDIX: DATASETS

## E.1 CIFAR-5M

CIFAR-5m is a dataset of 6 million synthetic CIFAR-10-like images. We release this dataset publicly on Google Cloud Storage, as described in `https://github.com/preetum/cifar5m`.

The images are RGB $32 \times 32$px. We generate samples from the Denoising Diffusion generative model of Ho et al. (2020) trained on the CIFAR-10 train set (Krizhevsky, 2009). We use the publicly available trained model and sampling code provided by the authors at `https://github.com/hojonathanho/diffusion`. We then label these unconditional samples by a 98.5% accurate Big-Transfer model (Kolesnikov et al., 2019). Specifically, we use the pre-trained BiT-M-R152x2 model, fine-tuned on CIFAR-10 using the author-provided code at `https://github.com/google-research/big_transfer`. We use 5 million images for training, and reserve the remaining images for the test set.

The distribution of CIFAR-5m is of course not identical to CIFAR-10, but is close for research purposes. For example, we show baselines of training a network on 50K samples of either dataset (CIFAR-5m, CIFAR-10), and testing on both datasets. Table 1 shows a ResNet18 trained with standard data-augmentation, and Table 2 shows a WideResNet28-10 (Zagoruyko and Komodakis, 2016) trained with cutout augmentation (DeVries and Taylor, 2017). Mean of 5 trials for all results. In particular, the WRN-28-10 trained on CIFAR-5m achieves 91.2% test accuracy on the original CIFAR-10 test set. We hope that as simulated 3D environments become more mature (e.g. Gan et al. (2020)), they will provide a source of realistic infinite datasets to use in such research.

Random samples from CIFAR-5m are shown in Figure 19. For comparison, we show random samples from CIFAR-10 in Figure 20.

| Trained On | Test Error On | |
|---|---|---|
| | CIFAR-10 | CIFAR-5m |
| **CIFAR-10** | 0.050 | 0.096 |
| **CIFAR-5m** | 0.110 | 0.106 |

Table 1: ResNet18 on CIFAR-10/5m

| Trained On | Test Error On | |
|---|---|---|
| | CIFAR-10 | CIFAR-5m |
| **CIFAR-10** | 0.032 | 0.091 |
| **CIFAR-5m** | 0.088 | 0.097 |

Table 2: WRN28-10 + cutout on CIFAR-10/5m

## E.2 IMAGENET: DOGBIRD

The ImageNet-DogBird task is constructed by collapsing classes from ImageNet. The task is to distinguish dogs from birds. The dogs are all ImageNet classes under the WordNet synset "hunting dog" (including 63 ImageNet classes) and birds are all classes under synset "bird" (including 59 ImageNet classes). This is a relatively easy task compared to full ImageNet: A ResNet-18 trained on 10K samples from ImageNet-DogBird, with standard ImageNet data augmentation, can achieve test accuracy 95%. The listing of the ImageNet wnids included in each class is provided below.

**Hunting Dogs (n2087122):** n02091831, n02097047, n02088364, n02094433, n02097658, n02089078, n02090622, n02095314, n02102040, n02097130, n02096051, n02098105, n02095889, n02100236, n02099267, n02102318, n02097474, n02090721, n02102973, n02095570, n02091635, n02099429, n02090379, n02094258, n02100583, n02092002, n02093428, n02098413, n02097298, n02093754, n02096177, n02091032, n02096437, n02088632, n02092339, n02099712, n02088632, n02093647, n02098286, n02096585, n02093991, n02100877, n02094114, n02101388, n02089973, n02088094, n02088466, n02093859, n02088238, n02102480, n02101556, n02089867, n02099601, n02102177, n02101006, n02091134, n02100735, n02099849, n02093256, n02097209, n02091467, n02091244, n02096294

**Birds (n1503061):** n01855672, n01560419, n02009229, n01614925, n01530575, n01798484, n02007558, n01860187, n01820546, n01817953, n01833805, n02058221, n01806567, n01558993, n02056570, n01797886, n02018207, n01828970, n02017213, n02006656, n01608432, n01818515, n02018795, n01622779, n01582220, n02013706, n01534433, n02027492, n02012849, n02051845, n01824575, n01616318, n02002556, n01819313, n01806143, n02033041, n01601694, n01843383, n02025239, n02002724, n01843065, n01514859, n01796340, n01855032, n01580077, n01807496, n01847000, n01532829, n01537544, n01531178, n02037110, n01514668, n02028035, n01795545, n01592084, n01518878, n01829413, n02009912, n02011460

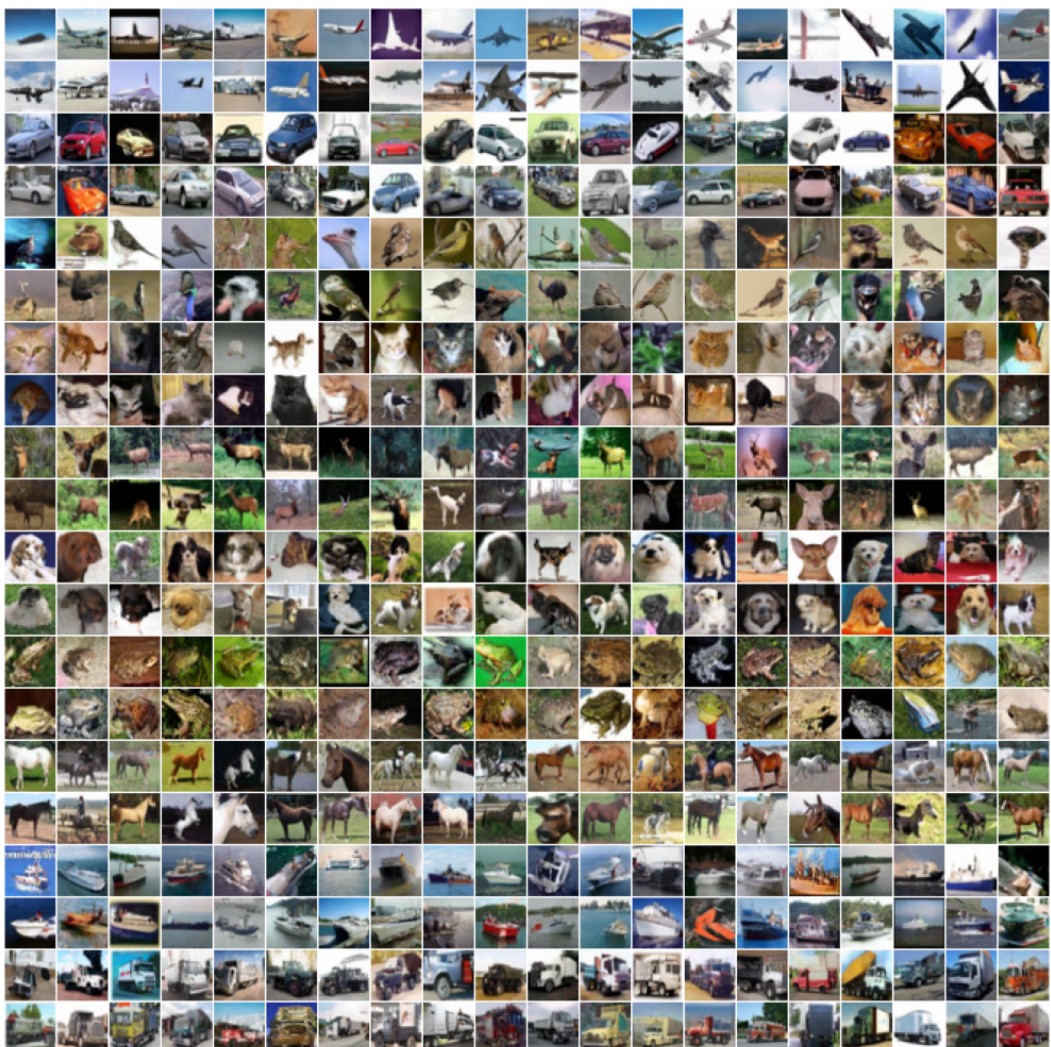

Figure 19: **CIFAR-5m Samples.** Random samples from each class (by row).

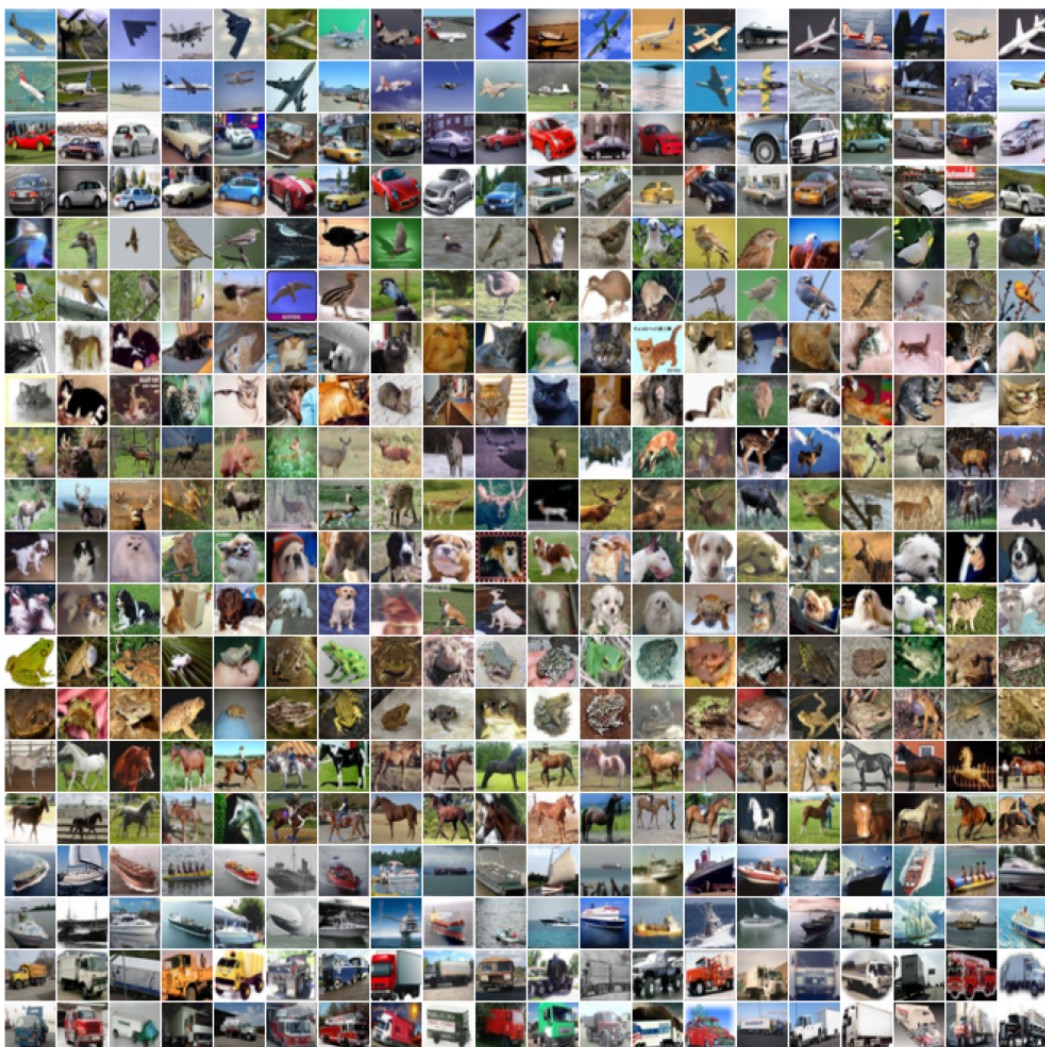

Figure 20: **CIFAR-10 Samples.** Random samples from each class (by row).

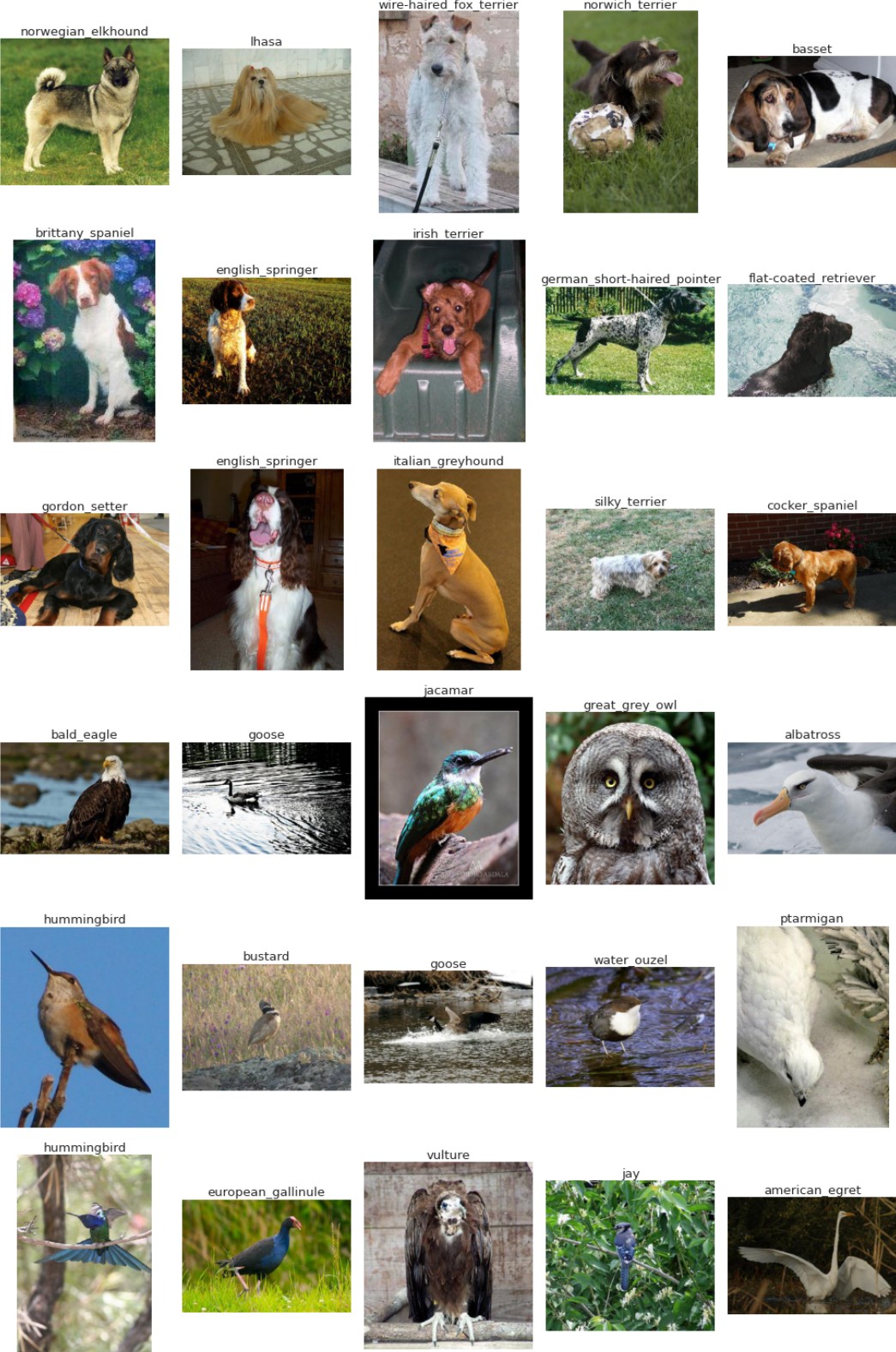

Figure 21: **ImageNet-DogBird Samples.** Random samples from each class. Annotated by their original ImageNet class for reference.

## F  D-CONV, D-FC ARCHITECTURE DETAILS

For convenience, we provide PyTorch specifications for the D-CONV and D-FC architectures from Neyshabur (2020) which we use in this work.

**D-CONV.** This model has 6563498 parameters.

```
Network(
  (features): Sequential(
    (0): Conv2d(3, 32, kernel_size=(3, 3), stride=(1, 1), padding=(1, 1)
, bias=False)
    (1): BatchNorm2d(32, eps=1e-05, momentum=0.1, affine=True)
    (2): ReLU(inplace=True)
    (3): Conv2d(32, 64, kernel_size=(3, 3), stride=(2, 2), padding=(1, 1)
, bias=False)
    (4): BatchNorm2d(64, eps=1e-05, momentum=0.1, affine=True)
    (5): ReLU(inplace=True)
    (6): Conv2d(64, 64, kernel_size=(3, 3), stride=(1, 1), padding=(1, 1)
, bias=False)
    (7): BatchNorm2d(64, eps=1e-05, momentum=0.1, affine=True)
    (8): ReLU(inplace=True)
    (9): Conv2d(64, 128, kernel_size=(3, 3), stride=(2, 2), padding=(1, 1)
, bias=False)
    (10): BatchNorm2d(128, eps=1e-05, momentum=0.1, affine=True)
    (11): ReLU(inplace=True)
    (12): Conv2d(128, 128, kernel_size=(3, 3), stride=(1, 1), padding=(1, 1)
, bias=False)
    (13): BatchNorm2d(128, eps=1e-05, momentum=0.1, affine=True)
    (14): ReLU(inplace=True)
    (15): Conv2d(128, 256, kernel_size=(3, 3), stride=(2, 2), padding=(1, 1)
, bias=False)
    (16): BatchNorm2d(256, eps=1e-05, momentum=0.1, affine=True)
    (17): ReLU(inplace=True)
    (18): Conv2d(256, 256, kernel_size=(3, 3), stride=(1, 1), padding=(1, 1)
, bias=False)
    (19): BatchNorm2d(256, eps=1e-05, momentum=0.1, affine=True)
    (20): ReLU(inplace=True)
    (21): Conv2d(256, 512, kernel_size=(3, 3), stride=(2, 2), padding=(1, 1)
, bias=False)
    (22): BatchNorm2d(512, eps=1e-05, momentum=0.1, affine=True)
    (23): ReLU(inplace=True)
  )
  (classifier): Sequential(
    (0): Linear(in_features=2048, out_features=2048
, bias=False)
    (1): BatchNorm1d(2048, eps=1e-05, momentum=0.1, affine=True)
    (2): ReLU(inplace=True)
    (3): Dropout(p=0.5, inplace=False)
    (4): Linear(in_features=2048, out_features=10, bias=True)
  )
)
```

**D-FC.** This model has 1170419722 parameters.

```
Network(
  (features): Sequential(
    (0): Linear(in_features=3072, out_features=32768, bias=False)
    (1): BatchNorm1d(32768, eps=1e-05, momentum=0.1, affine=True
    (2): ReLU(inplace=True)
    (3): Linear(in_features=32768, out_features=16384, bias=False)
```

```
    (4): BatchNorm1d(16384, eps=1e-05, momentum=0.1, affine=True)
    (5): ReLU(inplace=True)
    (6): Linear(in_features=16384, out_features=16384, bias=False)
    (7): BatchNorm1d(16384, eps=1e-05, momentum=0.1, affine=True)
    (8): ReLU(inplace=True)
    (9): Linear(in_features=16384, out_features=8192, bias=False)
    (10): BatchNorm1d(8192, eps=1e-05, momentum=0.1, affine=True)
    (11): ReLU(inplace=True)
    (12): Linear(in_features=8192, out_features=8192, bias=False)
    (13): BatchNorm1d(8192, eps=1e-05, momentum=0.1, affine=True)
    (14): ReLU(inplace=True)
    (15): Linear(in_features=8192, out_features=4096, bias=False)
    (16): BatchNorm1d(4096, eps=1e-05, momentum=0.1, affine=True)
    (17): ReLU(inplace=True)
    (18): Linear(in_features=4096, out_features=4096, bias=False)
    (19): BatchNorm1d(4096, eps=1e-05, momentum=0.1, affine=True)
    (20): ReLU(inplace=True)
    (21): Linear(in_features=4096, out_features=2048, bias=False)
    (22): BatchNorm1d(2048, eps=1e-05, momentum=0.1, affine=True)
    (23): ReLU(inplace=True)
  )
  (classifier): Sequential(
    (0): Linear(in_features=2048, out_features=2048, bias=False)
    (1): BatchNorm1d(2048, eps=1e-05, momentum=0.1, affine=True)
    (2): ReLU(inplace=True)
    (3): Dropout(p=0.5, inplace=False)
    (4): Linear(in_features=2048, out_features=10, bias=True)
  )
)
```

