# OpenReview forum: "The Deep Bootstrap Framework: Good Online Learners are Good Offline Generalizers"
_ICLR.cc/2021/Conference — ICLR 2021 Poster_

### Official Review · AnonReviewer4 · 2020-10-28
**Interesting empirical phenomenon**

**Rating:** 7
**Confidence:** 5

**Review:**

This paper studies generalization through a novel "bootstrap" framework relating the test loss from training on a finite fixed training set to the loss when training on fresh samples at each iteration. The framework is quite novel and suggests alternative perspectives on understanding empirical phenomena such as the success of overparametrization, data augmentation, and implicit biases.

Positives:
- The paper is clearly written
- The empirical bootstrap phenomenon is very interesting and the experimental results are quite surprising.

Cons:
- The paper could benefit from some more prominent discussion of settings where the bootstrap error is small vs large. It was briefly mentioned in this sentence in the conclusion: " The bootstrap error is not universally small for all models and
learning tasks: for example, we found the gap was larger at limited sample sizes and without data
augmentation. " Space permitting, it would be helpful and more convincing if more details were provided about settings where the bootstrap error could be large.
- The choice of soft error to compute the bootstrap error seems somewhat ad-hoc, especially for the datasets studied where the Bayes optimal risk is presumably quite low. For example, it seems like ImageNet-DogBird should be quite easy to separate unambiguously.

Some other comments and questions:
- It would be interesting to hear the authors thoughts on the feasibility of rigorously analyzing the bootstrap error. Are there any obstacles to analyzing this error that arise in the prior literature?

---

> ### Author Response · Authors · 2020-11-14
> **Response to R4**
>
> Thank you for your feedback, and for recognizing the empirical contribution of this work.
> We address your questions and concerns below.
>
> **“settings where the bootstrap error is small vs large”**: We agree that characterizing the dependency of the bootstrap error on problem parameters is an important question. We do not fully understand this yet, but we did include several investigations into this in the Appendix. For example, Appendix A shows that the bootstrap error can be either large or small in a linear regression setting, depending on whether the model is “misspecified.” Figure 10a in the Appendix also shows experiments without data-augmentation, which has a slightly larger bootstrap error than usual. In general, we found that the bootstrap error was small for “realistic” settings in deep learning (large enough ‘n’, standard data augmentation, deep neural networks) -- but it is an open question to understand what separates these “realistic” settings from “unrealistic” settings like linear regression. (We consider linear regression to be unrealistic here because it is not a deep network, and is not used in modern high-dimensional vision tasks in practice).
>
> **Regarding the choice of Soft-Error**:
>
> - While we found the Soft-Error to be better behaved than the Error, our bootstrap framework is still meaningful with respect to Test Error (which is the standard metric in classification). We included an additional plot in the revision to illustrate this -- please see Appendix B.6 and Figure 14, which shows the difference between Real and Ideal worlds with respect to error.
>
> - Proper choice of metric can be very important in studying deep learning. For example, it is known that Test Loss can be poorly behaved in classification settings, even when Test Error is well-behaved (e.g. our Figure 6, or Figure 3 in [1]). We thus believe that identifying Soft-Error as a metric is a contribution in itself -- we identify a meaningful metric for which standard-trained networks are very well-behaved, even if they are less well-behaved in other metrics.
>
> - As you note, we also have some “theoretically-inspired” motivations for choosing Soft-Error, based on distributional generalization of [Nakkiran & Bansal 2020]. Although the true Bayes optimal risk is 0 in our distributions, our intuition is that there is some “effective” risk due to the limitations of our learning procedures. That is, parts of the distribution that are “hard to learn” may look “pseudorandom” to our networks, and thus our networks may act as though the Bayes risk is non-zero (i.e. the “hard” parts look like “random noise”). This is somewhat handwavy, but it is our intuition for why Soft-Error is better-behaved than Error even when the true Bayes risk is 0.
>
> **Regarding “feasibility of rigorously analyzing the bootstrap error”**: the bootstrap error as we define it is closely related to quantities studied in the non-parametric bootstrap from statistics. However, in the statistics literature the bootstrap is analyzed for much simpler estimators than deep neural networks. Though we are not yet aware of direct routes to analyze the bootstrap error for neural nets, we hope our work will spur research into this direction (which has not been studied before, to the best of our knowledge). It may be easiest to first study the bootstrap error in simplified toy models, such as the linear regression setting in Appendix A.
>
> [1] “The Implicit Bias of Gradient Descent on Separable Data” Soudry et al. ICLR 2018. https://openreview.net/pdf?id=r1q7n9gAb

---

> > ### Comment · AnonReviewer4 · 2020-11-23
> > **Thanks for the response!**
> >
> > Thanks for the response, this answers my questions.

---

### Official Review · AnonReviewer2 · 2020-10-28
**Interesting framework but there appear to be some caveats**

**Rating:** 6
**Confidence:** 4

**Review:**

######################################################################

1.  Paper Summary

The authors propose a bootstrap framework for understanding generalization in deep learning.  In particular, instead of the usual decomposition of test error as training error plus the generalization gap, the bootstrap framework decomposes the empirical test error as online error plus the bootstrap error (the gap between the population and empirical error).  The authors then demonstrate empirically on variants of CIFAR10 and a subset of ImageNet that the bootstrap error is small on several common architectures.  Hence, the empirical test error is controlled by the online error (i.e. a rapid decrease in the error in the online setting leads to low test error).  The authors then provide empirical evidence to demonstrate that same techniques perform well in both over and under-parameterized regimes.

######################################################################

2. Strengths

2.1. The bootstrap framework presented for understanding generalization is novel to the best of my knowledge and provides an interesting connection between optimization in online learning and generalization in offline learning.

2.2. When the bootstrap error is low, the bootstrap framework implies that understanding generalization reduces to understanding optimization in the online setting.  This perspective provides an alternative to characterizing the implicit bias of deep networks when explaining generalization in deep networks.

2.3. The authors present significant empirical evidence that the bootstrap error (in terms of soft error) is consistently low in image classification settings across a number of architectures.

######################################################################


3. Concerns

3.1. While at first this framework serves as a very appealing alternative to the classical decomposition, the fact that the bootstrap error is low seems heavily reliant on the soft-error only found in the classification networks using a softmax on the outputs.  In particular, the authors focus on the soft-error in their decomposition throughout the work, but in Appendix A and in section 6, it appears that the MSE/cross entropy loss produce a significant bootstrap gap.  At first, this may seem innocuous, but if this bootstrap gap is low only for soft error, this would imply that we need to then focus on understanding how quickly the population soft error decreases, which to the best of my knowledge is a very different object of study in optimization than the MSE or cross entropy loss.

3.2.  In light of the previous point, I feel that the following experiment could help solidify whether the bootstrap error is indeed low for convolutional networks used for classification. When using the square loss for classification as is done in (https://arxiv.org/abs/2006.07322) (for example on the CIFAR10 setting), is the bootstrap error still low or is the bootstrap error being low really only a side effect of measuring soft-error? This experiment should resolve the authors' claim with the gap in Figure 6c being large due to unbounded weights when minimizing the empirical loss.

3.3. (Minor) I have a hard time understanding how strong some of the claims are in the paper.  In particular, the authors claim a main implication as "The same techniques [...] are used in practice in both over- and under-parameterized regimes."  Do the authors mean to claim that a technique that is successful in the over-parameterized regime is also successful in the under-parameterized regime? They do state this in briefly on page 6, but it seems to be claimed less strongly in the introduction.  Additionally, in light of the fact that there are cases where the bootstrap error is high, the claim that "the generalization of models in offline learning is largely determined by their optimization speed in online learning" may need to be more precise.

######################################################################

4. Score and Rationale

Overall, I currently vote for rejecting. However, my decision is borderline.  In particular, while I find the authors' bootstrap framework appealing in that it could nicely connect generalization and optimization, I am concerned about the results relying heavily on the soft-error (especially since the softmax activation is not necessary for deep networks to perform well on test data).  I am definitely open to changing my review provided that the authors are able to address my concerns about the soft-error above.

######################################################################


5. Questions

Please clarify the questions listed in the concerns section above.

######################################################################

6. Post-rebuttal Updates

In light of the authors new experiments and response, I have increased my score.  The bootstrap error shows slight deviations under the MSE, and thus, I do feel that the scope of generality of this work is still limited.  However, I feel that the presented framework is novel and the experiments consistently demonstrate that the bootstrap error is consistently low under soft-error in a number of settings.

---

> ### Author Response · Authors · 2020-11-14
> **Response to R2**
>
> Thank you for your detailed and valuable feedback! We are glad your review recognizes the novelty and significance of our contribution. To resolve your concern about the soft-error, we ran several experiments in the last few days results of which are added to the revision and will be used as evidence and referred to in our response below. We hope that you would consider increasing your score to “accept” if your concerns are adequately addressed.
>
> **Regarding the choice of Soft-Error**:
>
> - While we found the Soft-Error to be better behaved than the (classification) Error, our bootstrap framework is still meaningful with respect to Test Error (which is the standard metric in classification). We included an additional plot in the revision to illustrate this -- please see Appendix B.6 and Figure 14, which shows the difference between Real and Ideal worlds with respect to error.
>
> - Following your suggestion, we performed experiments on CIFAR-10 networks trained with squared loss (without a softmax layer, as in [Hui & Belkin 2020]). In these settings, the bootstrap gap (with respect to Test Error) was still reasonably small, though not as small as for standard networks trained with cross-entropy. Please see Appendix B.6.1 and Figure 15 for these additional experiments.
>
> - Finally, proper choice of metric can be very important in studying deep learning. For example, it is known that Test Loss can be poorly behaved in classification settings, even when Test Error is well-behaved. That is, if we only looked at the Test Loss, we may mistakenly conclude that networks are overfitting, even when the Test Error does not overfit (e.g. our Figure 6, moreover, this point has been made in many prior works such as Figure 3 in [1]). We thus believe that identifying Soft-Error as a metric is a contribution in itself -- we identify a meaningful metric for which standard-trained networks are very well-behaved, even if they are poorly-behaved in other metrics.
>
> **Regarding the minor concerns**:
> - To clarify the strength of our various claims: In this work, Claim 1 is the main and strongest claim: that the bootstrap error is small for “realistic” settings in deep learning. The purpose of Section 4 is to establish and test this claim across various realistic settings. Once we are convinced that the Real World is often close to the Ideal World (i.e. Claim 1), we can use this perspective as a lens on other phenomena in deep learning. This is what we do in Section 5. That is, the experiments and discussion in Section 5 show how to use our framework to gain insight into many other aspects of deep learning. The claims in Section 5 are, as you note, not as precise and thorough as our main Claim -- this is because we preferred to showcase many connections to other areas instead of focusing on only a few connections.
>
> - The statement “The same techniques [...] are used in practice in both over- and under-parameterized regimes” is meant to be only an observation about the current state of ML practice. This statement should be somewhat surprising a-priori (since, why would these two regimes be connected?), but our bootstrap perspective gives a consistent way of thinking about this statement. Specifically, our bootstrap framework suggests that we would expect models which work well in the under-parameterized regime (i.e. good online learners) to also work well in the over-parameterized regime (i.e. good offline generalizers).
>
> - Regarding “cases where the bootstrap error is high”: In our experiments, the cases with high bootstrap error were somewhat “unrealistic” -- either small number of samples, or unrealistic models (e.g. the linear models in Appendix A, which are theoretically tractable but not used in modern high-dimensional vision tasks). Thus, we believe our main claim (Claim 1) holds for all “realistic” settings and practical scenarios , as specified.
>
> - That said, we agree that some of our statements in the Introduction are slightly imprecise, since we have not yet introduced the full formalism required to state them precisely. In this case, we chose to include these informal statements in the abstract / introduction because they convey the essence of the idea concisely, and they are turned into more precise claims later. However, we understand if such statements in the “Our Contributions” section are confusing, and we can consider removing them from the Introduction. We appreciate your thoughts on the tradeoffs involved.
>
> [1] “The Implicit Bias of Gradient Descent on Separable Data” Soudry et al. ICLR 2018. https://openreview.net/pdf?id=r1q7n9gAb

---

> > ### Comment · AnonReviewer2 · 2020-11-23
> > **Follow up**
> >
> > Thank you for performing the experiments on the square loss and adding in the additional figure to the appendix.
> >
> > * Regarding your points about loss vs error
> >     * I definitely expected to see a difference between the two when considering the cross entropy loss due to the increasing norm of the parameters, but I’m a bit surprised by the emergence of a gap in the case of MSE loss (especially on resnets). I feel that overall, the identification of soft-error is interesting, but it limits the scope of generality in this work.   In particular, since networks trained using MSE still generalize although the bootstrap error demonstrates deviations in a few cases.  Regardless, I do feel that the framework is novel and the bootstrap error under soft-error is consistently small across a number of models.  Hence, I will increase my score slightly.
> >     * (Minor) I’m sorry for the delay in my response, and so I understand if the following experiment cannot be done in time, but does the deviation for MSE get more pronounced with even deeper resnets?
> >
> > * Regarding the strength of claims
> >     * Thank you for clarifying about the main claims in this work.  My personal suggestion would be to strengthen the claim regarding “the same techniques […],” and connect it with your finding (i.e. explaining how the bootstrap error unifies the principles used in the two regimes as on page 6).  Without further clarification, this will likely not come across as surprising to readers as stated.  For example, we almost always use the same optimization techniques for over and under-parameterized models, but the surprising aspect is the benefit of the technique.  This is more clearly explained on page 6 (section 5), and I feel that this could be better emphasized in the introduction.

---

> > > ### Author Response · Authors · 2020-11-24
> > > **Response to R2 follow-up**
> > >
> > > Thank you for engaging with our response and increasing your score.
> > > We like your suggestion about strengthening and elaborating on claims in the Introduction, and we will incorporate it in the final version.
> > > We can also perform more MSE-experiments with deeper resnets in the final version.
> > >
> > > Regarding your comment on the “limited scope of generality”: We believe that this scope is quite broad and relevant, since training with cross-entropy and softmax activation is standard procedure in many practical works (including works achieving SOTA). While we agree that it would be desirable to have a universal theory that explains all possible neural network settings (incl training with MSE), we believe it is already quite general to include only softmax/cross-entropy networks.
> > > Note that we do test a variety of architectures in this setting, including transformers (Image-GPT), which further shows the generality of our results.
> > >
> > > Moreover, our framework is still relevant even under MSE, though as you note there are slight deviations. We hope that future work can extend our framework to see both MSE-networks and softmax-networks in a unified way (perhaps by defining the “right” notion of soft-error for MSE-networks).

---

### Official Review · AnonReviewer3 · 2020-10-31
**a concern about the novelty and practicality of the proposed generalization gap framework**

**Rating:** 4
**Confidence:** 3

**Review:**

This paper defines a metric of generalization gap between the ideal world and real-world via 'bootstrap approximation', and seeks to use his gap to explain some phenomena.

Although I have respect for this paper trying to define some statistical terms in the deep learning society, I have a very strong concern about the novelty, as it appears to be whether trivial or hard to understand. For example, I could relate to the empirical finding that bootstrap error defined in claim 1 is uniformly small, but I personally believe that the 'TestSoftError' defined in 3 should dominate the generalization error and the paper circumvents this term to study some term that is way non-significant, especially considering how the datasets are built around CIFAR or DOG-bird, via simple data augmentation that is frequently used in vision society. The problem being mentioned in the paper, i.e. 'Most if not all techniques for understanding the generalization gap (e.g. uniform convergence, VC-dimension, regularization, stability, margins) remain vacuous', appears to be not tackled at all in this paper, which, in its main effort, to discuss generalization.

Section 5 says some practical suggestions to train deep networks, which seem to be not persuasive, especially that the ideal world error might be harder to measure than that in the real world, according to the paper. On top of this, why these factors such as sample size, model selection, are not to be tuned even better if that were through the lens of other metrics, such as gradient norms, validation error, etc.

---

> ### Author Response · Authors · 2020-11-14
> **Response to R3 (part 1/2)**
>
> Thank you for your feedback. Your main concern appears to be that our results seem trivial. To clarify this, we elaborate on the motivation and context of our results, and explain why they are far from trivial. Taking the following context into account, if you find our contributions to be significant, we appreciate it if you increase your score to acceptance. Otherwise, we appreciate it if you could comment on the points of disagreement.
>
> First, we summarize the main contributions and mention our understanding of reviewers’ stance of that (please let us know if our understanding is not correct):
> 1. We propose a decomposition of Test Error into two terms: (A) The “Ideal World” test error, and (B) the “bootstrap error.”: The reviewer has not discussed this.
> 2. We argue, through extensive experiments, that the second term (B) is uniformly small in realistic settings: The Reviewer agrees with this claim.
> 3. This reduces studying test performance to studying term (A): The reviewer seems unsatisfied that we have not understood term (A), and instead focused only on establishing term (B).
>
> We now clarify the contribution in each of these points.
>
> **Step 1** is novel: This decomposition has not been proposed before, and suggests an interesting connection between online and offline learning. Of course, there are many possible decompositions of test error, but only some of them are scientifically interesting (please see the introduction for the discussion of the significance). This brings us to the second step.
>
> **Step 2** is surprising and fundamental: It is a-priori far from obvious that the bootstrap error should be small, especially for networks which nearly fit their train sets. For example, in Figure 1 the Real World sees 50K unique samples, while the Ideal World sees 5 million unique samples. And yet, these two worlds surprisingly have similar test performance.
> (Regarding your comment on data-augmentation: The bootstrap error is still reasonably small even without data-aug. See Figures 10a and 10b in the Appendix).
>
> Perhaps the most surprising part about Step 2 is that the bootstrap error is **uniformly** small. That is, it is small regardless of whether the network is a CNN, an MLP, a Transformer, a ResNet, etc. This suggests that the bootstrap error is a fundamental object in deep learning, and not something specific to our particular design choices.
>
> Finally, even if the Reviewer is unsurprised by this experimental observation: it is not something that has been demonstrated before, and not to be taken as granted -- thus, it requires the rigorous experimental validation presented in our work to verify.
>
>
> **Step 3** lays the foundation of a new research program.
> The reviewer is correct that we currently only understand term B, the “bootstrap error”, and not term A, the “Ideal World test error.”
> If we fully understood both terms, we would have a completely solved generalization in deep learning, which is an ambitious goal, and not one we claim in this paper.
>
> Rather, the purpose of our paper is to direct the focus of the generalization research community into studying term (A). This is a significant divergence from current approaches, because our work argues that “optimization is all you need” to study generalization. Notice that the “Ideal World Test Error” is a problem in online optimization, which does not involve generalization.
>
> This is in contrast to many prevailing approaches to studying generalization, which argue that optimization is **insufficient** to study generalization. For example, see the survey post by Sanjeev Arora [ http://www.offconvex.org/2019/06/03/trajectories/ ], and many works on “implicit bias” on deep learning [ https://arxiv.org/abs/1412.6614 and subsequent works]. All of these works suggest that optimization considerations alone cannot characterize generalization, but our work suggests the opposite.
>
> In summary, the goal of our paper is not to completely understand generalization, but to outline a promising new research program to understand generalization. We show a surprising connection between offline and online optimization, which reduces understanding generalization to understanding **online optimization.** The reviewer correctly points out that we do not fully understand online optimization (term B), but we hope that our paper inspires future research in this direction.

---

> > ### Author Response · Authors · 2020-11-14
> > **Response to R3 (part 2/2)**
> >
> > **Regarding Section 5**: The point of Section 5 is not to directly provide practical suggestions on tuning neural networks, but rather to provide insight into various phenomena in deep learning. To clarify, the aim of our paper is to help build a better scientific understanding of generalization, and not directly to improve current methods. This is in a similar vein as papers such as “Understanding deep learning requires rethinking generalization” [Zhang et al. ICLR 2017], which inspired the community and moved the needle forward through various follow up works. We propose our alternative approach to understanding generalization that has the potential to lead to many more subsequent insights, some of which we include in Section 5.

---

### Official Review · AnonReviewer1 · 2020-11-06
**New Framework, but Not Good Enough**

**Rating:** 5
**Confidence:** 3

**Review:**

This paper proposes a bootstrap framework to study the generalization problem of deep learning, by decomposing the traditional test error into an ‘Ideal World’ test error plus the gap between. Empirically, it demonstrates that such gap (soft-error) is small in supervised image classification for typical deep learning model architectures. It then proposes to explain several phenomena in deep learning using the bootstrap framework. I appreciate the authors' efforts on conducting extensive experiments related to deep learning generalization.

Pros:

+ The proposed decomposition of test error is new to the field.

+ The presented experiments are extensive, which connected to various important aspects in deep learning.


Cons:

- My main concern is that the motivation of the proposed bootstrap framework is unclear. I have a hard time understanding how the proposed framework overcomes the two major obstacles of classical approaches listed in the introduction section. Why are the subproblems presented in (2) easier than the original one?

- Classical generalization bounds (although can be vacuous in some scenarios) are mostly theoretically-derived, which are able to characterize the dependencies of the error term on parameters, such as input dimension, sample size and model class complexities. However, the main claim of the paper (claim 1 in Page 3) is a pure experimental claim. The paper will be much stronger, if claim 1 can be characterized theoretically with bounds on the error term (even in a simplified setting).

- The implications of the proposed framework and results on how to improve the existing deep learning methods are not clear to me. Assuming the bootstrap error is always small, then how can one modify the current deep learning training method to reduce the ‘Ideal World’ test error?

- Characterizing the ‘Ideal World’ test error requires a significant large amount of labeled inputs for supervised image classifications. The paper construct synthetic datasets by first generating images using an unconditional generative model, then labeling the images using a pretrained classification model. How do you guarantee the label quality for the whole generated dataset? Why not directly use a conditional generative model?

Minor Comments:

1. The mathematical definitions of test error and test soft error are not provided. I would recommend the authors to lay out these metrics in the main body of the paper to add clarity.


===== Post-Discussion Update =====

I appreciate the authors' efforts for responding my concerns and comments. The provided response and the update of the introduction do help better explain the motivations and the implications of the proposed generalization framework. Therefore, I raise my previous rating a little bit to reflect this.

Although the proposed framework may suggests a potentially-interesting future direction for the deep learning research community, it still does not fully convince me its feasibility. In particular, the paper would be much stronger, if the author can dig deeper with the Real-World test (soft) error, which I view it as the end-goal of a real-world classification system, to provide more specific directions on how to make it smaller, or more importantly, how it changes the current deep learning training paradigm.

---

> ### Author Response · Authors · 2020-11-14
> **Response to R1**
>
> Thanks for your valuable feedback. We appreciate you recognizing the novelty of our proposal, and the thoroughness of our experiments. Below, we answer the questions raised by the reviewer. We hope that given the reviewer’s positive view on the novelty and quality of the experiments, they increase the score to “accept” if they are satisfied with the answers to their questions.
>
>
> **1- Motivation**:
> Regarding your main concern about the motivation: The main reason we believe our proposed decomposition in eq. (2) is promising is not because it is a decomposition into two easier pieces, but into two *different* and *familiar* pieces. That is, for interpolating classifiers, the original decomposition in eq. (1) amounts to writing test error as 0 + {test error}, which is not a useful decomposition since one of the pieces is 0.
>
> However, our decomposition in eq. (2) writes test error as two *different* objects directly related to online optimization and bootstrap error in optimization theory and statistics.
> These  tools have not previously been applied to study generalization in deep learning (as we describe on page 2).
>
> Given the difficulties encountered by many decades of work on the original decomposition (1), we believe proposing this alternate decomposition is significant as it creates the opportunity to benefit from tools in optimization theory and statistics to theoretically study generalization. (Whether this route will be ultimately successful remains to be seen, in future work).
> We further provide empirical evidence that the bootstrap error remains small which is an important hint towards attacking the problem by showing that perhaps the generalization mystery can be understood by understanding deep learning in the online setting.
>
> To address your remarks, we have updated the Introduction of our paper to more clearly explain this motivation.
>
>
> **2- “Pure Experimental Claim”**:
> This is indeed a purely experimental paper -- however, this is not a shortcoming.
> The first step in scientific study is often to identify and establish an empirical phenomena/conjecture. Once the empirical behavior is understood, it can help guide a deeper theoretical understanding. For example, the implicit regularization in matrix factorization was first conjectured in [1] and was supported with experimental evidence and some analysis for a very limited case. However, that encouraged many researchers to work on that direction and various version of the conjecture was proved, eg. see [2,3].
> Our work is the first step, and serves to introduce the novel idea which is non-trivial and establish it experimentally. We put considerable effort into experimental design to validate our conjecture empirically. We hope that it will inspire further theoretical study.
>
> Finally, in Appendix A, we explore certain toy linear regression models, in order to isolate the important aspects of our experiments that may be transferable to theoretically-tractable settings.
>
>
> **3- “How to improve deep learning methods”**:
> To clarify, the aim of our paper is to help build a better scientific understanding of generalization, and not directly to improve current methods.
> This is in a similar vein as papers such as “Understanding deep learning requires rethinking generalization” [Zhang et al. ICLR 2017], which inspired the community and moved the needle forward through various follow up works. We propose our alternative approach to understanding generalization that has the potential to lead to many more subsequent insights.
>
>
> **4- Synthetic dataset**:
> Regarding the synthetic dataset, we used an unconditional model because at the time this work was performed, the unconditional DDPM model appeared to have the highest-quality samples among CIFAR-10 generative models.
>
> Moreover, note that we have a number of non-synthetic experiments as well (ImageNet-DogBird, and even an original CIFAR-10 experiment in Figure 5b).
>
>
> **Re. Minor comment**:
> We have made the definition of soft-error more prominent in the revision (Section 2).
>
>
> We hope that these comments address your concerns. Please let us know if you have any remaining concerns/comments.
>
> [1] Suriya Gunasekar, Blake E Woodworth, Srinadh Bhojanapalli, Behnam Neyshabur, and Nati Srebro. Implicit regularization in matrix factorization. NIPS 2017.
>
> [2] Li, Yuanzhi, Tengyu Ma, and Hongyang Zhang. "Algorithmic regularization in over-parameterized matrix sensing and neural networks with quadratic activations." COLT, 2018.
>
> [3] Sanjeev Arora, Nadav Cohen, Wei Hu, Yuping Luo. Implicit Regularization in Deep Matrix Factorization. NeurIPS 2019.

---

### Comment · ~Shaohua_Li2 · 2021-03-12
**Possible concerns about the use of synthetic images**

This is an interesting work. Though, I find one thing that's concerning: the authors used a GAN to generate 5 million images to approximate the ideal world. However, similar attempts of using BigGAN generated images to train a classifier have led to worse or equal performance as using the real images [1,2]. A possible explanation is that, although the synthetic images look diverse for humans, they may actually lie in low-dimensional manifolds for which a CNN may overfit easily. Therefore I'm not very sure whether GAN-generated images constitute a good approximation of the real image distribution of the ideal world.

[1] Suman Ravuri and Oriol Vinyals, "Seeing is not necessarily believing: Limitations of biggans for data augmentation," 2019.

[2] Victor Besnier, Himalaya Jain, Andrei Bursuc, Matthieu Cord, Patrick Pérez, "This dataset does not exist: training models from generated images," arXiv:1911.02888, 2019.

---

> ### Comment · ~Preetum_Nakkiran1 · 2021-03-12
> **Response to Shaohua Li**
>
> Hi Shaohua,
>
> Note that we also have ImageNet experiments, with entirely real data (non-synthetic). See Section 4, and in particular Figure 3. This should address your concern with synthetic images.
>
> Regarding how realistic our synthetic image distributions are: Tables 1 and 2 in the appendix include some baselines. For example, training a WRN28-10 on 50K samples from CIFAR-5m reaches 91.2% test accuracy on the *original* CIFAR-10 test set.

---

### Comment · ~Léon_Bottou1 · 2021-03-23
**Pointing out a related work...**


This feels very close to some of my old work, which itself builds on classic arguments from the sixties on the asymptotic efficiency of online algorithms. Basically one pass of second order stochastic gradient (hence online because it only sees each example only once) matches the test performance of the empirical optimum (on the same data).

See the slides of the talk and the paper links :  https://leon.bottou.org/talks/onepass

---

> ### Comment · ~Hanie_Sedghi1 · 2021-03-30
> **Response to Leon Bottou**
>
> Dear Leon, thank you for bringing these phenomenal works to our attention -- we will add them in the revised version.
> As you have mentioned, our work is similar to yours in nature. You showed that (under certain assumptions), second order stochastic optimization converges towards the minimum of the expected cost just as fast as any batch algorithm.
> We consider a different setting as detailed below, but it’s interesting that the results have a similar flavor.
>
> Specifically, the main differences in our setting are:
> -We study non-convex models, and consider the entire optimization trajectory (from initialization, not just in "final stage" of training).
> -Our results compare the *same* first-order optimization algorithm in both online and offline settings.
> -We describe non-asymptotic behaviors (at finite time, and finite sample sizes).
> -Finally, we focus on empirical results for neural-networks used in practice, but we do not have corresponding theoretical results.
>
> That said, the results in your works are very related, and it would be great to extend those theoretical results to the setting in our work.

---

### Decision · Program_Chairs · 2021-01-07
**Final Decision**

**Decision:**

Accept (Poster)

**Comment:**

The paper is proposing a new framework for understanding generalization in the deep learning. The main idea is considering the difference of stochastic optimization on a population risk and optimization on an empirical risk. The classical theory considers the difference of empirical risk and population risk. This basically translates the practical motivation from finding good function classes to finding good optimizers which can re-use the data effectively. Although the paper provides no theoretical result, it provides an interesting empirical study. The paper somewhat demonstrates that SGD on deep networks is somehow good at re-using the same data. I believe this angle is very novel and might hope to future theoretical discoveries. The paper is reviewed by four reviewers and two of them argue its acceptance and two of them argue rejection. After discussion, this status remained and I carefully read and reviewed the paper. Here are the major issues raised by the reviewers:

- R#1: The paper is missing a theoretical study. The implications on the practical deep learning is not clear.
- R#2: Choice of the soft-error is particular to the task and how to go beyond soft-max is not clear.
- R#3: Finds the paper not novel as well as trivial or hard to understand.
- R#4: The choice of soft error is ad-hoc.

I believe the issues raised by R#3 are not justified. First of all, novelty is very clear and. appreciated by other reviewers. Moreover, the paper is rather easy to understand and the results are very farm from trivial. However, the other issues raised by other reviewers are valid. Specifically, soft-error seems to be a limitation of the study. However, the authors respond to this concern and reviewer increases their score. I believe the theory is lacking but the paper is simply showing this novel approach and its empirical validity. A theory to explain this phenomenon would be amazing but not necessary for publication. Similarly, without a theory it is hard to expect any practical implication. Overall, I believe the paper is an interesting and novel one which will likely to lead additional work in the area. Considering we are still far from a satisfying theory of generalization for deep learning and the role of the optimization is clear, this angle worth sharing with the community. Hence, I decide to accept. However, I have some concerns which should be addressed by the camera-ready.

- Claims should be revised and authors should make sure they have enough evidence for them. For example, authors provide no satisfying evidence for random labels or very limited evidence for pre-training. I strongly recommend authors to either remove some of these discussions or present in a fashion which is not a result but part of the discussion for future research.
- A section about limitations should be added. Specifically, the soft-error choice should be discussed in this limitation section.
- Discussion section should be extended with the pointers to the relevant work on bootstrap literature as well as suggestions to the theoreticians. Not providing any theoretical result is always fine but authors should understanding why is it hard to make theoretical statements and where to search them.